# Heterogeneous impacts of HIV pre-exposure prophylaxis (PrEP) on drug resistance and phylogenetic cluster transmission dynamics in British Columbia, Canada: A retrospective cohort and simulation study

Angela McLaughlin[1,2], Junine Toy[1], Vincent Montoya[1], Paul Sereda[1], Jason Trigg[1], Mark Hull[1], Chanson J. Brumme[1,3], Rolando Barrios[1], Julio S. G. Montaner[1,3], Jeffrey B. Joy[1,2,3]*

1 British Columbia Centre for Excellence in HIV/AIDS, Vancouver, Canada, 2 Bioinformatics, University of British Columbia, Vancouver, Canada, 3 Department of Medicine, University of British Columbia, Vancouver, Canada

* jbjoy@mail.ubc.ca

## Abstract

### Background

HIV pre-exposure prophylaxis (PrEP) prevents infection when used during periods of risk, however, its population-level effectiveness is hindered by incomplete uptake, adherence, and retention. Since oral PrEP became available free-of-cost in British Columbia (BC), Canada, in January 2018, uptake has been rapid among eligible individuals, primarily comprising gay, bisexual, and other men who have sex with men (GBM), however, its effectiveness against HIV acquisition across subpopulations alongside potential effects on baseline drug resistance have not been estimated. We evaluated individual and population-level impacts of PrEP on HIV drug resistance and transmission in phylogenetic clusters, representing groups of individuals linked by recent outbreaks, to elucidate heterogeneity in its effectiveness.

### Methods and findings

Using a retrospective cohort design, we evaluated the frequencies of baseline drug resistance mutations and membership in phylogenetic clusters among newly HIV diagnosed people who ever filled a prescription for HIV PrEP (i.e., PrEP users) in BC ($n = 39$) compared to non-PrEP users ($n = 566$) diagnosed from 2018 to 2022 in the BC Drug Treatment Program with at least one sequence available. Newly HIV diagnosed PrEP users were significantly more likely than newly diagnosed non-PrEP users to be included in phylogenetic clusters (chi-squared test, $p = 0.0075$) and carry baseline nucleoside analogue reverse transcriptase inhibitor (NRTI) resistance

**Data availability statement:** The British Columbia Centre for Excellence in HIV/AIDS is prohibited from making individual-level data available publicly owing to provisions in our service contracts, institutional policy, and ethical requirements. To facilitate research, we make such data available via data access requests. Some data from the Centre are not available externally owing to prohibitions in service contracts with our funders or data providers. Institutional policies stipulate that all external data requests require collaboration with a researcher from the British Columbia Centre for Excellence in HIV/AIDS. For more information or to make a request, please contact Mark Helberg (mhelberg@bccfe.ca). Source code, tables, and figures are available at github.com/angmcl/hiv-prep-bc.

**Funding:** AM was supported by a Canada Graduate Scholarship Doctoral award from the Canadian Institutes of Health Research (CIHR) https://cihr-irsc.gc.ca/e/193.html and the BC-CfE https://bccfe.ca/. JSGM was supported by grants paid to his institution, the BC-CfE, by the BC Ministry of Health https://www2.gov.bc.ca/gov/content/governments/organization-al-structure/ministries-organizations/ministries/health, Health Canada https://www.canada.ca/en/health-canada.html, Public Health Agency of Canada (PHAC) https://www.canada.ca/en/public-health.html, Vancouver Coastal Health https://www.vch.ca/en, Vancouver General Hospital Foundation https://vghfoundation.ca/, and Genome BC https://www.genomebc.ca/. JBJ was supported by CIHR, Genome BC GeneSolve Grant (GEN054), Genome Canada https://genomecanada.ca/, PHAC, and the BC-CfE. VM, PS, JT, MH, CJB, and RB were also supported by the BC-CfE. The BC-CfE contributed to data collection, entry, storage, processing, and querying. Other funders had no role in study design, data collection and analysis, decision to publish, or preparation of the manuscript.

**Competing interests:** We have read the journal's policy and the authors of this manuscript have the following competing interests: JSGM received institutional grants from Gilead

mutation M184I/V (Fisher's exact test, adjusted $p$-value = 0.025). Subsequently, we quantified the population-level impacts of widespread PrEP availability on transmission based on the effective reproduction number ($R_e$), compared across key populations living with HIV in BC and active phylogenetic clusters with at least one new case since 2018. We applied simulations of active clusters' growth based on their empirically observed $R_e$ with or without estimated PrEP impacts to estimate diagnoses averted via PrEP across clusters, with non-clustered cases grouped together. Most diagnoses were averted in large and medium GBM-predominant clusters. In a Poisson model, clusters with fewer diagnoses averted were associated with having a higher median age and lower proportion of new diagnoses with PrEP use, adjusted for cluster size at the end of 2017 and proportion residing in Vancouver Coastal Health Authority. These results must be interpreted in light of uncertainty owing to incomplete sampling, the use of consensus genomes, phylogenetic inference, and the assumptions of counterfactual simulations.

## Conclusions

We estimated that the oral PrEP program in BC from 2018 to 2022 averted approximately 20 new HIV diagnoses per year across phylogenetic clusters, while infrequently contributing to baseline drug resistance in instances where PrEP was inadvertently prescribed during acute infection or with incomplete adherence. These findings corroborate the broad effectiveness of PrEP, describe heterogeneity in its impacts on clusters' growth, and suggest groups for prioritized PrEP services.

---

Author summary
## Why was this study done?

- HIV pre-exposure prophylaxis (PrEP) prevents HIV acquisition, but poses the risk of selecting for drug resistance mutations if taken inadvertently during acute infection or with low adherence, and may not be equitably effective.

- Since it became widely available free-of-cost to eligible individuals, primarily in the gay, bisexual, and other men who have sex with men population, in British Columbia, Canada in January 2018, PrEP's effectiveness at reducing new HIV cases across different subpopulations of the epidemic has not been elucidated.

## What did the researchers do and find?

- A retrospective cohort study was designed to compare drug resistance mutations and linkage to active outbreaks, identified through phylogenetic clusters, comprising 39 newly HIV diagnosed people who ever filled a prescription for HIV PrEP (i.e., PrEP users) from 2018 to 2022, compared with 566 newly diagnosed non-PrEP users.

Sciences, Merck, and ViiV Healthcare, which had no financial stakes in the results of the current study. CJB has received grants and honoraria as speaker and consultant paid to his institution from Merck, Gilead Sciences, and ViiV Healthcare, which also had no stakes in the results of the study. MH acted as site investigator for Gilead clinical trial for HIV therapeutics. All other authors declare no competing interests.

**Abbreviations:** ART, antiretroviral therapy; BC, British Columbia; CI, confidence interval; DTP, Drug Treatment Program; FTC, emtricitabine; GTR, generalized time reversible; HIVdb, HIV Drug Resistance Database; ML, maximum likelihood; NRTI, nucleoside analogue reverse transcriptase inhibitor; PLWH, people living with HIV; PrEP, pre-exposure prophylaxis; RAMs, resistance-associated mutations; $R_e$, effective reproduction number; SDRMs, surveillance drug resistance mutations; TasP, Treatment as Prevention; TDF, tenofovir disoproxil fumarate; TGM, transgender men; VL, viral load.

- Newly HIV diagnosed PrEP users were significantly more likely than non-PrEP users to have acquired baseline mutation M184I/V, conferring resistance to antiretroviral drugs present in PrEP and some antiretroviral therapy regimens. Previous PrEP users were also more likely to be members of clusters, representing local outbreaks.

- We compared heterogeneity in clusters' transmission activity, estimated through the effective reproduction number ($R_e$), since widespread PrEP in 2018, and then used observed dynamics to fit simulations of clusters' growth in the absence of PrEP to estimate how many cases may have been averted in each cluster.

- Across clusters, we estimate that PrEP averted about 20 new HIV infections annually from 2018 to 2022, but it was most effective in reducing $R_e$ and averting new cases in medium and large gay, bisexual, and other men who have sex with men (GBM) clusters. In a Poisson model, fewer cases were averted in clusters comprised of older individuals with infrequent previous PrEP use.

### What do these findings mean?

- These findings corroborate the importance of monitoring both HIV resistance-associated mutations, including among those with previous PrEP use, as well as phylogenetic clusters, which differ in their underlying subpopulations and transmission dynamics in the presence of treatment and prevention interventions.

- Limitations in interpreting this study include incomplete sampling of people living with HIV within and outside of BC, sensitivity to cluster definition including a minimum size, and uncertainty in estimating transmission due to sparse case counts and individual-level variation.

### Introduction

Optimal control of HIV/AIDS requires equitable delivery of antiretroviral therapy (ART) to people living with HIV (PLWH) immediately following HIV diagnosis, and prevention interventions, including pre-exposure prophylaxis (PrEP), to populations at elevated risk. Provision of prioritized treatment and prevention resources can be informed by epidemiological monitoring, including via identification of HIV phylogenetic clusters, representing linkage through recent transmission. Although HIV morbidity, mortality, and incidence have declined in many regions, including in British Columbia (BC), Canada [1], ongoing transmission continues to disproportionately affect certain populations.

Oral PrEP effectively prevents HIV-1 acquisition when taken during periods of risk among gay, bisexual, and other men who have sex with men (GBM) [2], transgender women (TGW) [2], heterosexuals (HET) [3], and people who inject drugs (PWID) [4]. Barriers to access, racial disparities [5], limited eligibility or coverage, inconsistent adherence [6], and program non-retention [7] have resulted in sub-optimal oral PrEP effectiveness. In 2016, Health Canada approved once-daily oral PrEP, Truvada,

comprised of tenofovir disoproxil fumarate (TDF)/emtricitabine (FTC) for adults at elevated risk of HIV acquisition, including GBM and serodiscordant heterosexual couples [8], followed by generic TDF/FTC in 2017 [9].

Since January 1, 2018, PrEP has been fully publicly funded and centrally distributed for BC residents with greater likelihood of HIV acquisition as determined by local PrEP guidelines [10]. PrEP uptake in BC has expanded rapidly, with 9,737 cumulative participants by June 30, 2022 [7,11], consisting of 97.0% cisgender GBM, 1.3% TGW, 0.9% cisgender women, and 0.5% transgender men (TGM) [7]. Those who were not cisgender men (i.e., TGM, TGW, and cisgender women) were less likely to persist with PrEP, as were those who were younger and had no prior PrEP use [7]. Differences in PrEP awareness, uptake, access, and persistence manifest as differential reductions in HIV transmission. Lapsed prescriptions (greater than six months beyond expected refill date based on daily PrEP use) and incomplete adherence have resulted in 39 newly HIV diagnosed people who ever filled a prescription for HIV PrEP (i.e., PrEP users) between October 23, 2018 and December 5, 2022 (0.4% of PrEP participants).

Population-level effectiveness of PrEP has been estimated through cohort studies, simulations, and mathematical models, but not at the phylogenetic cluster level. In a compartmental model of HIV in BC, prioritized provision of PrEP to GBM with higher likelihood of HIV acquisition, in combination with Treatment as Prevention (TasP), was associated with reduced incidence and effective reproduction number ($R_e$) [12,13]. Grouping the BC population by HIV exposure, testing, and PrEP interest clarified that the association between behavior and PrEP interest improved its population-level effectiveness [14]. In other settings, PrEP availability resulted in a 25% relative reduction of new HIV diagnoses in GBM in an Australian cohort [15], and a stochastic mechanistic model suggested PrEP averted 20% of HIV acquisitions from 2015 to 2021 in Montreal [16]. Expanding PrEP coverage to 80% was estimated to avert 8% of new HIV infections in Australia, 15% in Thailand, and 26% in China over 40 years [17], and in the United States, 33% of cases over the next decade could be averted with 40% coverage of MSM and 62% adherence [18]. While studies have consistently found PrEP effective, the heterogeneity of those impacts across subpopulations has been inadequately considered.

Pre-treatment drug resistance originating from transmission of resistant variants or acquired resistance due to the use of TDF/FTC during acute HIV infection or before diagnoses, adversely impacts clinical outcomes by limiting treatment options [19–21]. Mutations in *reverse transcriptase* (*rt*) conferring resistance to nucleoside analogue reverse transcriptase inhibitor (NRTI) drugs found in PrEP and ART, TDF (K65R) and FTC (M184I/V), were identified in individuals retrospectively found to have acute HIV infections who were prescribed PrEP [22]. In New York, past and recent PrEP use were associated with significantly elevated risk of baseline M184I/V, adjusted for transmission risk group [19]. In a meta-analysis, individuals exposed to PrEP during acute infection were significantly more likely to have TDF or FTC resistance [23]. HIV acquisition during PrEP use has occurred in randomized control trials, but is confounded by incomplete adherence and reporting bias [22]. Evaluating HIV drug resistance in phylogenetic trees can help to distinguish emergence and transmission of resistance-associated mutations (RAMs) within phylogenetic clusters routinely monitored for HIV surveillance and research in BC [24–29].

In this study, we sought to estimate impacts of PrEP use on baseline drug resistance and phylogenetic clustering among new HIV diagnoses, and the population-level impacts of PrEP on the $R_e$ of phylogenetic clusters to inform the effectiveness of PrEP in averting new cases and to advocate for prioritized provision of PrEP to subpopulations with cluster growth potential. We tested the hypotheses that newly HIV diagnosed PrEP users were more likely to be included in BC phylogenetic clusters than newly diagnosed non-PrEP users generally and among GBM, under the expectation that PrEP eligibility and use is related to risk behavior and network connectivity, and that this results in an elevated likelihood of linkage to active outbreaks, and that PrEP users were more likely than non-PrEP users to have baseline NRTI drug resistance mutations due to inadvertent PrEP use during acute infection. We further hypothesized that $R_e$ and diagnoses averted have been impacted heterogeneously across phylogenetic clusters since PrEP availability, based on the expectation of differential PrEP awareness, access, and retention in subpopulations of social networks represented as clusters, amid broad $R_e$ reductions in the BC GBM population.

## Methods

### Study setting and data source

HIV sequences and patient metadata for this study were obtained on February 9, 2023, representing data up to December 31, 2022, from the Drug Treatment Program (DTP) at the BC Centre for Excellence in HIV/AIDS (BC-CfE) representing all PLWH connected to publicly-funded treatment in BC, Canada. This study includes 14,919 DTP participants, of which 10,740 had at least one HIV partial *pol* sequence available. De-identified participant metadata included gender (male; TGM; female; TGF), sex-at-birth, age, residence census tract and regional health authority, prescriber census tract, date of first viral load (VL) test, date of first ART, ART prescriptions, PrEP dispensations (if applicable), self-reported risk exposures at enrollment (GBM; PWID; HET; blood; hepatitis C virus; other), VL over time, baseline CD4+ T-cell (CD4) count, prior acquired immune deficiency syndrome (AIDS) diagnosis, and if applicable, date of death. The retrospective cohort of newly diagnosed PrEP users and non-PrEP users included a subset of DTP participants described below.

Sequencing was performed by clinical staff at the BC-CfE and genotypic data were stored in the BC-CfE access-controlled facility within a secure, encrypted Oracle database. Investigators had no access to nominal data. As this study was based on analysis of doubly-anonymized viral sequence data, consent was waived for this study. Research ethics were approved by University of British Columbia Providence Health Care Research Institute (REB #H20-02859) on December 7, 2020 and amended for an updated data request on September 8, 2022. The ethics protocol and application were included in S3 File. The final study differed from the prospective analysis plan in regards to the method applied to estimate $R_e$, for which we initially proposed using Bayesian phylodynamic birth-death skyline models [30] and instead used EpiEstim to estimate instantaneous $R_e$ [31]. This change followed the discovery that birth-death(-sampling) models are non-identifiable in the absence of additional data [32,33]. The other change from the initial protocol was the inclusion of stochastic simulations of clusters' growth with and without PrEP based on the observed $R_e$ dynamics and in consideration of the impact of both PrEP and COVID-19 on HIV transmission.

### Phylogenetic and drug resistance analyses

We analyzed 41,941 HIV-1 partial *pol* (*protease (pro)* and partial *rt*) sequences from 10,740 DTP participants collected between May 30, 1996 and December 31, 2022 and 8,739 *integrase* (*int*) sequences from 4,052 patients (1−35 sequences per patient). Additional details on data cleaning were included in S1 Text. Partial *pol* and *int* sequences were aligned to the HXB2 reference genome (GenBank Accession #K03455) using minimap2 parallelized by viralMSA [34,35]. COMET was used to assign HIV-1 subtypes using partial *pol* [36]. For COMET subtype assignments with less than 90% bootstrap support, subtype assignment was evaluated using REGA [37], and the assignment with higher support was chosen. Sequences from newly HIV diagnosed individuals were screened for RAMs using the Stanford HIV Drug Resistance Database (HIVdb) version 9.6 updated March 9, 2024 [38]. Codons of surveillance drug resistance mutations (SDRMs) [39], and insertions relative to reference HXB2 were removed prior to phylogenetic inference.

A set of 100 bootstrap nucleotide alignments were generated to infer bootstrap approximate maximum likelihood (ML) phylogenetic trees in FastTree 2.1 with a generalized time reversible (GTR) substitution model [40]. Trees were outgroup rooted on the oldest subtype B sequence. Phylogenetic clusters were identified as groups of at least five individuals whose viruses share a pairwise patristic (tree) distance <0.02 substitutions per site in greater than 90% of bootstraps [24]. This pairwise tree distance threshold was identified as the 95th percentile of within-host pairwise distances for subtype B in BC [24] and is used for phylogenetic monitoring in BC. We evaluated the effect of rooting using outgroups, midpoint, and root-to-tip regression on the distribution of cluster sizes (Fig C in S1 Text). ML trees were inferred for key clusters with IQ-TREE 2 under a GTR substitution model with 1,000 resamples for ultrafast bootstrap support values [41,42], then rooted using root-to-tip regression using sample collection dates optimized by residual mean squares in ape (Figs G–O in S1 Text) [43].

## Statistical comparison of newly HIV diagnosed with and without previous PrEP use

To evaluate individual-level impacts of PrEP on baseline drug resistance and clustering, we applied a retrospective cohort study design to compare newly HIV diagnosed previous PrEP users ($n = 39$) with date of first detectable VL from Oct 23, 2018 to Dec 5, 2022, with a control group of newly diagnosed non-PrEP users ($n = 566$) over the same period. Cohort eligibility included DTP participants with at least one sequence available and a first detectable VL from Oct 23, 2018 to Dec 5, 2022, excluding those with previous ART experience (PLWH who have immigrated). Individuals enrolled for PrEP who never filled prescriptions were considered non-PrEP users. This component of the study was reported as per the Strengthening the Reporting of Observational Studies in Epidemiology (STROBE) guidelines (S2 STROBE checklist).

Characteristics associated with PrEP use among newly HIV diagnosed (Table C in S1 Text) and characteristics associated with clustering stratified by PrEP use (Table D in S1 Text) were evaluated with chi-squared tests for categorical variables and Kruskal-Wallis tests of medians for numeric variables. We tested the hypotheses that newly diagnosed PrEP users included a higher proportion of clustered individuals and higher proportion of baseline NRTI resistance than non-PrEP users overall and among GBM. Chi-squared tests were applied if greater than 80% of expected frequencies exceeded five, otherwise Fisher's exact tests were applied. To corroborate differences in clustering, lineage-level viral diversification rates in newly diagnosed with or without PrEP were compared using Kruskal-Wallis tests (Fig B in S1 Text). Proportions of newly diagnosed with and without PrEP use who had baseline drug resistance mutations were compared using Fisher's exact tests with $p$-values adjusted using the Benjamini–Hochberg method (Table 2) [44]; statistical power was calculated for the M184IV Fisher's test using alpha 0.05.

## Effective reproduction number dynamics since PrEP

Diagnoses over time were used to estimate incidence [45] and instantaneous $R_e$ using EpiEstim with gamma-distributed serial intervals [31] for BC, key populations (GBM, PWID, HET), and 29 active phylogenetic clusters (with at least one new diagnosis since 2018) and at least 10 members. Parameters compared for $R_e$ estimation in BC (Figs S–V in S1 Text), key populations (Figs W and X in S1 Text), and clusters (Figs Y–AB in S1 Text) included a gamma-distributed serial interval with mean of 0.5 year (y), 1 y, 2 y, or 5 y, and standard deviation (sd) of 0.5 y, 1 y, or 2 y; estimation interval of 0.25 y, 0.5 y, or 1 y; and $R_e$ smoothing over 30 days (d), 90 d, or 365 d. These serial interval ranges reflect that a substantial proportion of transmission occurs during early infection due to a high VL, with infrequent longer intervals for transmission from chronic infections [46,47]. Further, 36% of HIV infections in the US from 2004 to 2008 were diagnosed within 1 y [48]. Primary analyses represent a serial interval with mean 1 y, sd 0.5 y, estimation interval 0.5 y, and 90 d $R_e$ smoothing. We fitted average $R_e$ in the periods before PrEP (Jan 2016–Dec 2017), during PrEP and before the coronavirus disease 2019 (COVID-19) pandemic (Jan 2018–Feb 2020), during PrEP and COVID-19 (Mar 2020–Feb 2022), and during PrEP and post-COVID-19 (Mar 2022–Dec 2022). Fold-change in $R_e$ with PrEP (PrEP effect) was calculated as the ratio of $R_e$ during PrEP and before COVID-19 versus $R_e$ before PrEP in key populations and clusters (i.e., PrEP effect < 1 means a reduction of $R_e$).

## Stochastic simulations of phylogenetic cluster growth with and without PrEP

To reflect infrequent superspreading and variable generation times, stochastic branching processes were used to simulate growth of active HIV clusters in BC with and without PrEP availability from 2018 to 2022, whereby the number of secondary cases for each infected case in each generation is drawn from a negative binomial distribution defined by mean of $R_e$ and dispersion, k [49–52]. $R_e$ was specified as either the observed cluster $R_e$ with PrEP or counterfactual cluster $R_e$ without PrEP adjusted by the PrEP effect (Fig AD in S1 Text). Observed cluster $R_e$ with 95% confidence interval (CI) width greater than 10 (sparse data) or not available were replaced with $R_e = 0.8$ to reflect inactivity. If PrEP effect could not be calculated for a cluster, either because it was a new cluster seeded since 2018 or $R_e$ could not be calculated before

or after PrEP, then the PrEP effect of the clusters' predominant population was assigned. For each generation, $R_e$ was permitted to deviate from the specified value by drawing from a binomial distribution (probability of success 50%), where $R_e$ was then drawn from a gamma-distribution centered on cluster $R_e$ (sd = 0.1). We assumed a gamma-distributed serial interval (mean 1 y, sd 0.5 y) and k sampled uniformly between 0.1 and 0.3 for each generation. The simulation parameters specified and estimated were reported with ranges of uncertainty (Table G in S1 Text) in accordance with guidelines for reporting simulations and mathematical models [53].

The seed size is defined as the number of infectious individuals at time zero (January 1, 2018) for each active cluster (Table F in S1 Text). Clusters' initial seeds were estimated as total cases minus deaths and emigrations by the end of 2017, multiplied by the proportion expected to be virally unsuppressed based on estimates that 82% of cases in BC were diagnosed, 76% of those diagnosed were on ART, and 83% of those on ART were virally suppressed [54]. Following a simulation based on the initial seed, an adjustment factor for the seed was calculated as the ratio of observed new cases in each cluster from 2018 to end of 2022, with the mean number of sampled new cases from simulations using observed cluster $R_e$ (Table F, Fig V in S1 Text). The adjusted seed size was applied in subsequent simulations with both observed $R_e$ (with PrEP available) and counterfactual $R_e$ (PrEP absent).

For scenarios with or without PrEP, 4,000 simulations were run per cluster and for all non-clustered cases grouped as pseudo-cluster 9,999. Simulations began on day zero, January 1, 2018 and ran to January 1, 2023 (1,827 d). Infected cases resulting from simulations were stochastically diagnosed with probability sampled from a log-normal distribution centered on 0.82, and diagnosed cases were sampled (i.e., sequenced) with probability drawn from a log-normal distribution centered on 0.95 (~95% of DTP participants since 2018 have a *pol* sequence). CIs of diagnoses and samples averted were calculated by sampling 95% of simulations without removal 1,000 times, then calculating normally-distributed CIs across bootstrap means. Cluster diagnoses averted (i.e., cases averted) is calculated as the difference in means of cumulative diagnoses in simulations with and without PrEP (Figs W and X in S1 Text). Poisson generalized linear models were evaluated to identify cluster characteristics associated with the most and fewest diagnoses averted (Table H in S1 Text). Diagnoses or samples averted were shifted up by the minimum averted to remove negative integers. Factors evaluated included cluster size in 2017, predominant and percentage key populations, percentage health authority of residence, median member age in 2023 and age at first ART, and percentage of new diagnoses with PrEP experience.

## Results

### Elevated clustering of newly HIV diagnosed PrEP users compared to non-PrEP users

Over the course of 4.1 years from Oct 23, 2018 to Dec 5, 2022, newly HIV diagnosed PrEP users ($n = 39$) in BC were significantly more likely than newly diagnosed non-PrEP users ($n = 566$) to be male (100% versus 79.5%; chi-squared test, $p = 0.011$; Table C in S1 Text) and GBM (97.1% versus 56.7% of reported, $p < 0.001$), reflecting PrEP eligibility criteria, as well as younger at first ART (median 32 versus 37 y old; Kruskal–Wallis test, $p = 0.033$), and have a higher baseline CD4+ T-cell count (490 versus 380, $p = 0.003$).

By December 31, 2022, there were 246 BC HIV phylogenetic clusters, comprising at least five individuals whose viruses share pairwise patristic (tree) distance less than 0.02 substitutions per site in at least 90% of bootstraps, 84 of which were active, with at least one new diagnosis since 2018. Tree inference method negligibly affected cluster identification (Fig C in S1 Text). All clusters that included PrEP users also included at least one non-PrEP user (Fig A in S1 Text).

Newly HIV diagnosed PrEP users were significantly more likely to cluster than non-PrEP users in the entire cohort (chi-squared test $p = 0.0075$, Fig B in S1 Text), and among GBM ($p = 0.005$). Of 39 newly diagnosed PrEP users, 30 were a part of 13 active clusters (76.9% clustered, one-group proportions test 95% CI: 60.3, 88.3%), compared to 303 of 566 non-PrEP users (53.5% clustered, 49.3, 57.7%), who were a part of 84 active clusters (Table 1). Among those identifying as GBM, 28/34 (82.4%, 64.8, 92.6%) of newly diagnosed PrEP users were a part of phylogenetic clusters, compared to

Table 1. Phylogenetic clustering among newly HIV diagnosed in BC from 2015 to 2022, stratified by PrEP use from 2018 to 2022. The total newly diagnosed (*N*), total and percentage clustered, and unique clusters (*n*). Retrospective cohort was restricted to newly diagnosed with date of first detectable VL from October 23, 2018 to December 5, 2022.

| | All newly HIV diagnosed | | | | Newly HIV diagnosed, no PrEP | | | | Newly HIV diagnosed, PrEP | | | |
|---|---|---|---|---|---|---|---|---|---|---|---|---|
| Year | *N* | *N* clustered | % clustered | *n* clusters | *N* | *N* clustered | % clustered | *n* clusters | *N* | *N* clustered | % clustered | *n* clusters |
| 2015 | 256 | 140 | 54.7 | 48 | | | | | | | | |
| 2016 | 252 | 121 | 48.0 | 45 | | | | | | | | |
| 2017 | 183 | 103 | 56.3 | 43 | | | | | | | | |
| 2018 | 174 | 94 | 54.0 | 40 | 173 | 93 | 53.8 | 40 | 1 | 1 | 100.0 | 1 |
| 2019 | 168 | 83 | 49.4 | 43 | 160 | 78 | 48.8 | 42 | 8 | 5 | 62.5 | 4 |
| 2020 | 127 | 83 | 65.4 | 30 | 121 | 78 | 64.5 | 30 | 6 | 5 | 83.3 | 5 |
| 2021 | 131 | 74 | 56.5 | 34 | 125 | 70 | 56.0 | 34 | 6 | 4 | 66.7 | 3 |
| 2022 | 124 | 67 | 54.0 | 24 | 106 | 52 | 49.1 | 24 | 18 | 15 | 83.3 | 7 |
| **Sum *N* or Mean %, 2018–2022** | | | | | 685 | 371 | 54.2 | 84 | 39 | 30 | 76.9 | 13 |

145/265 (54.7%, 48.5, 60.8%) of non-PrEP users (Table D in S1 Text; chi-squared test $p = 0.005$). Newly diagnosed PrEP users also had significantly higher lineage-level viral diversification rates, indicative of rapid, recently sampled transmission events, than non-PrEP users (Fig B in S1 Text). In a logistic regression model, newly diagnosed PrEP users had 2.6-times (95% CI: 1.2, 6.4) higher odds of clustering compared to non-PrEP users, adjusted for GBM, age at first ART, and baseline CD4 count. The nine non-clustered newly diagnosed previous PrEP users (difference between total and n clustered in Table D in S1 Text) included a heterosexual male, and two minority subtypes A1 and 02_AG.

### Previous PrEP use associated with elevated probability of baseline NRTI drug resistance mutation M184I/V

Drug resistance mutations were identified and resistance scores were calculated using Stanford HIVdb version 9.6 [39]. Differences in prevalence of baseline RAMs between newly HIV diagnosed PrEP users and non-PrEP users were evaluated by drug class (Fig E in S1 Text) and mutation (Fig 1, Table 2). By drug class, there were no significant differences in the proportion of newly diagnosed PrEP users and non-PrEP users with baseline RAMs to any drug class (chi-squared test, $p = 0.12$), or to specific drug classes (NNRTI, Fisher's exact test: $p = 0.12$; NRTI, $p = 0.26$; PI, $p = 0.2$ (Fig E in S1 Text).

Among 39 newly diagnosed PrEP users, eight had baseline NRTI RAMs. Baseline NRTI resistance mutation M184I/V was more frequent in newly diagnosed PrEP users than non-PrEP users (Fig 1): 3 of 39 newly diagnosed PrEP users (7.7%) had baseline M184V or M184IV (referred to collectively as M184I/V), compared to 2 of 566 (0.35%) non-PrEP users (Fisher's exact test, adjusted $p$-value = 0.025; Table 2; power = 79% at alpha = 0.05). Elevated frequency of M184IV conferred elevated resistance scores to FTC, lamivudine (3TC), and abacavir (ABC) (Fig F in S1 Text). M184I/V was the only baseline RAM significantly more common in PrEP users than non-PrEP users (Table 2). Newly diagnosed PrEP users with baseline M184V had the two lowest proportion of days covered by PrEP (0.60 and 0.63 versus median = 0.88 across newly diagnosed PrEP users) and had significantly shorter time from last PrEP dispensation to first detectable VL (76 d versus 373 d; Kruskal–Wallis test, $p = 0.045$). The newly diagnosed PrEP user with M184IV had a PrEP prescription filled 29 days before first detectable VL. This was identified to be a diagnostic false negative, as the individual had acute HIV seroconversion at PrEP initiation and was diagnosed at 1 month follow-up. Nearest phylogenetic neighbors to viruses with M184I/V sampled from five newly diagnosed with and without prior PrEP use did not have detectable M184I/V (Fig 1B), suggesting it was unlikely to have been transmitted between hosts in any cases, but rather acquired *de novo*.

### Transmission of resistance-associated mutations in phylogenetic clusters

We evaluated whether and which baseline drug resistance mutations were found at high frequencies within clusters, suggestive of onward transmission in active outbreaks. There were 28 active clusters with at least one new diagnosis since

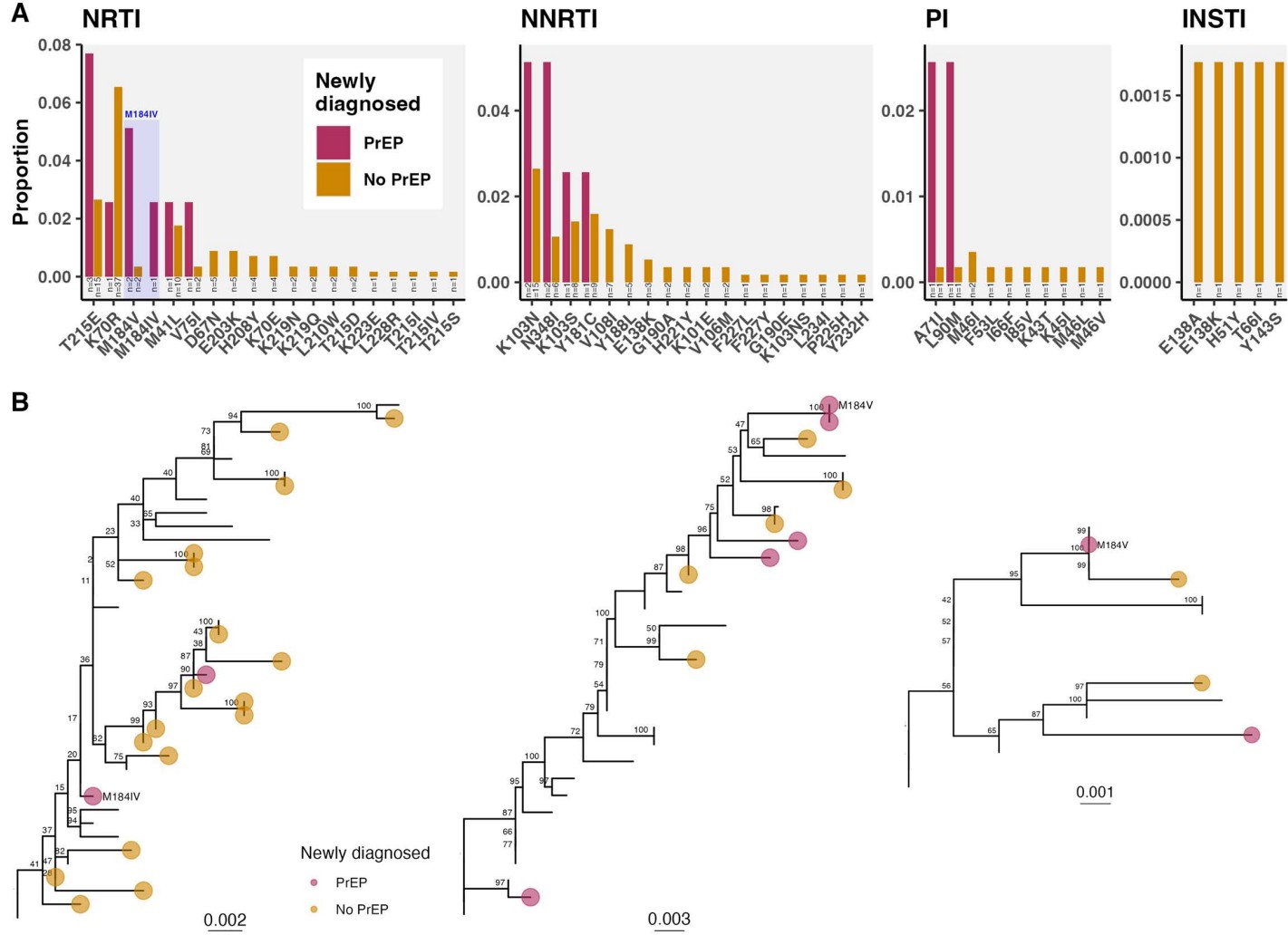

**Fig 1. Baseline HIV drug resistance in newly HIV diagnosed individuals with and without previous PrEP use. A)** Proportion of newly diagnosed with or without PrEP use with baseline drug resistance mutations to NRTI, NNRTI, PI, and integrase strand transfer inhibitor (INSTI) drug classes. Annotations indicate number of individuals with baseline resistance. Mutation M184I/V shaded in blue. **B)** Clustered viruses with baseline M184V or M184IV. Subtrees were pruned from maximum likelihood trees inferred for clusters including only the oldest sequence available per individual in IQ-TREE 2 and annotated with ultrafast bootstrap support values.

2018 with baseline resistance mutations (Fig 2). NRTI RAM, K70R, and NNRTI RAM, K103N, were increasingly detected at baseline (Fig D in S1 Text) and transmitted within clusters (Fig 2, Fig L in S1 Text). Several NRTI RAMs were detected multiple times within clusters, including thymidine analogue mutation revertant T215E (100% of 14 new cases in cluster 114, Fig L in S1 Text), K70R (27.3% of 11 new cases in cluster 49, Fig I in S1 Text; 100% of 2 new cases in cluster 185; and outside clusters, Fig 2), D67N (11.5% of 26 new cases in cluster 137, Fig M in S1 Text), and M41L (4.1% of 94 new cases in cluster 31, Fig H in S1 Text; present in clade of cluster 137, Fig M in S1 Text). Baseline NNRTI RAMs detected multiple times within clusters included K103N (2.1% of 94 new cases in cluster 31, Fig H in S1 Text; 80% of 5 new cases in cluster 97; 40% of 5 in cluster 194), K103S (83.3% of 6 new cases in cluster 234), Y181C (100% of 5 new cases in cluster 197), and N348I (85.7% of 7 new cases in cluster 41). No baseline INSTI RAMs were detected among newly diagnosed PrEP users, however, five non-PrEP users carried baseline INSTI RAMs E138A/K, H51Y, T66I, and Y143S (Fig 1).

**Table 2. Proportion of newly HIV diagnosed PrEP users and non-PrEP users 2018–2022 with resistance-associated mutations in genes, *rt* and *pro*.** Frequencies were compared using Fisher's exact tests and *p*-values were adjusted using the Benjamini–Hochberg method. M184V and M184I/V (including M184IV) are emboldened, and * denotes a *p*-value lower than significance level of 0.05.

| Drug class | Mutation | Gene | Newly HIV diagnosed PrEP users (*n*=39) | | Newly HIV diagnosed Non-PrEP users (*n*=566) | | Fisher's test | |
|---|---|---|---|---|---|---|---|---|
| | | | Total | % | Total | % | Unadj. *p* | Adj. *p* |
| NRTI | M41L | *rt* | 1 | 2.56 | 10 | 1.77 | 0.5226 | 0.5226 |
| | K70R | *rt* | 1 | 2.56 | 37 | 6.54 | 0.5012 | 0.5226 |
| | V75I | *rt* | 1 | 2.56 | 2 | 0.35 | 0.1815 | 0.3327 |
| | **M184V** | ***rt*** | **2** | **5.13** | **2** | **0.35** | **0.0224*** | **0.2463** |
| | **M184I/V** | ***rt*** | **3** | **7.69** | **2** | **0.35** | **0.0023*** | **0.0250*** |
| | T215E | *rt* | 3 | 7.69 | 15 | 2.65 | 0.1031 | 0.2747 |
| Non-NRTI (NNRTI) | K103N | *rt* | 2 | 5.13 | 15 | 2.65 | 0.3007 | 0.4725 |
| | K103S | *rt* | 1 | 2.56 | 8 | 1.41 | 0.4533 | 0.5226 |
| | Y181C | *rt* | 1 | 2.56 | 9 | 1.59 | 0.4891 | 0.5226 |
| | N348I | *rt* | 2 | 5.13 | 6 | 1.06 | 0.0886 | 0.2747 |
| Protease inhibitors (PI) | A71I | *pro* | 1 | 2.56 | 1 | 0.18 | 0.1249 | 0.2747 |
| | L90M | *pro* | 1 | 2.56 | 1 | 0.18 | 0.1249 | 0.2747 |

## Reduction of HIV reproduction number following PrEP among GBM

Within the DTP, HIV incidence declined from 9.2 new cases per 100,000 in 2012 to 2.7 new cases per 100,000 in 2022 (Fig O in S1 Text). Before PrEP, new HIV cases overall decreased annually by −12.1% in 2016 and −23.4% in 2017 (Fig 3), with most rapid declines among PWID (−19.7% in 2016, −49.0% in 2017), but less so among GBM (−4.7% in 2016, −11.4% in 2017, Fig Q in S1 Text). Following PrEP, annual new HIV cases in GBM declined −19.2% in 2018 and −8.9% in 2019, and GBM accounted for fewer new cases. During COVID-19, new HIV cases in GBM declined −26.1% in 2020, whereas they increased 6.4% among HET. Post-COVID, new HIV cases increased by 5.7% in GBM in 2022 likely due to increased potential exposures and testing, despite lowering −54.1% in PWID and −28.3% in HET. Among GBM, average $R_e$ was around 1 before PrEP, then dropped below 1 during PrEP, and further below 1 during COVID-19 (Fig 3, Fig V in S1 Text). $R_e$ among PWID (Fig W in S1 Text) and HET were elevated following PrEP, increased during COVID-19, and have been relatively low post-COVID-19.

## Differential cluster growth following availability of PrEP

We evaluated heterogeneity in $R_e$ dynamics in phylogenetic clusters following PrEP. There were 84 active clusters with at least one new case since 2018; 53 clusters had more than 10 members by Feb 2023 (Figs A and C in S1 Text). Of active clusters, 52 were predominantly (i.e., comprising the largest proportion of any risk group) GBM, 30 PWID, and 2 HET (Table E in S1 Text).

Transmission within and outside BC HIV phylogenetic clusters was variably impacted by PrEP (Fig 4, Figs X–AD in S1 Text). Cluster 31, the largest GBM-predominant cluster (size=391, 89% GBM, 94 new cases since 2018), had stable $R_e$ near 1, followed by an increase in mid-2017, then a decline since PrEP availability (fold-change $R_e$ with PrEP<1). Other GBM-predominant clusters (cluster 95, size=158, 27 new cases; and cluster 22, size=42, 19 new cases) had sustained declines through PrEP, with recent resurgences post-COVID-19 (Fig AC in S1 Text). The two largest PWID-predominant clusters (clusters 49 and 57, sizes 459 and 471, 91% and 86% PWID) have had limited new cases since 2018 (11 and 19). $R_e$ of PWID cluster 57 was stably below or near 1 from 2015 to 2018, then rose significantly above 1 in 2019, coming down in 2020 before rising above 1 post-COVID-19. PWID cluster 137 (size=90, 26 new cases, 88% PWID) has also had

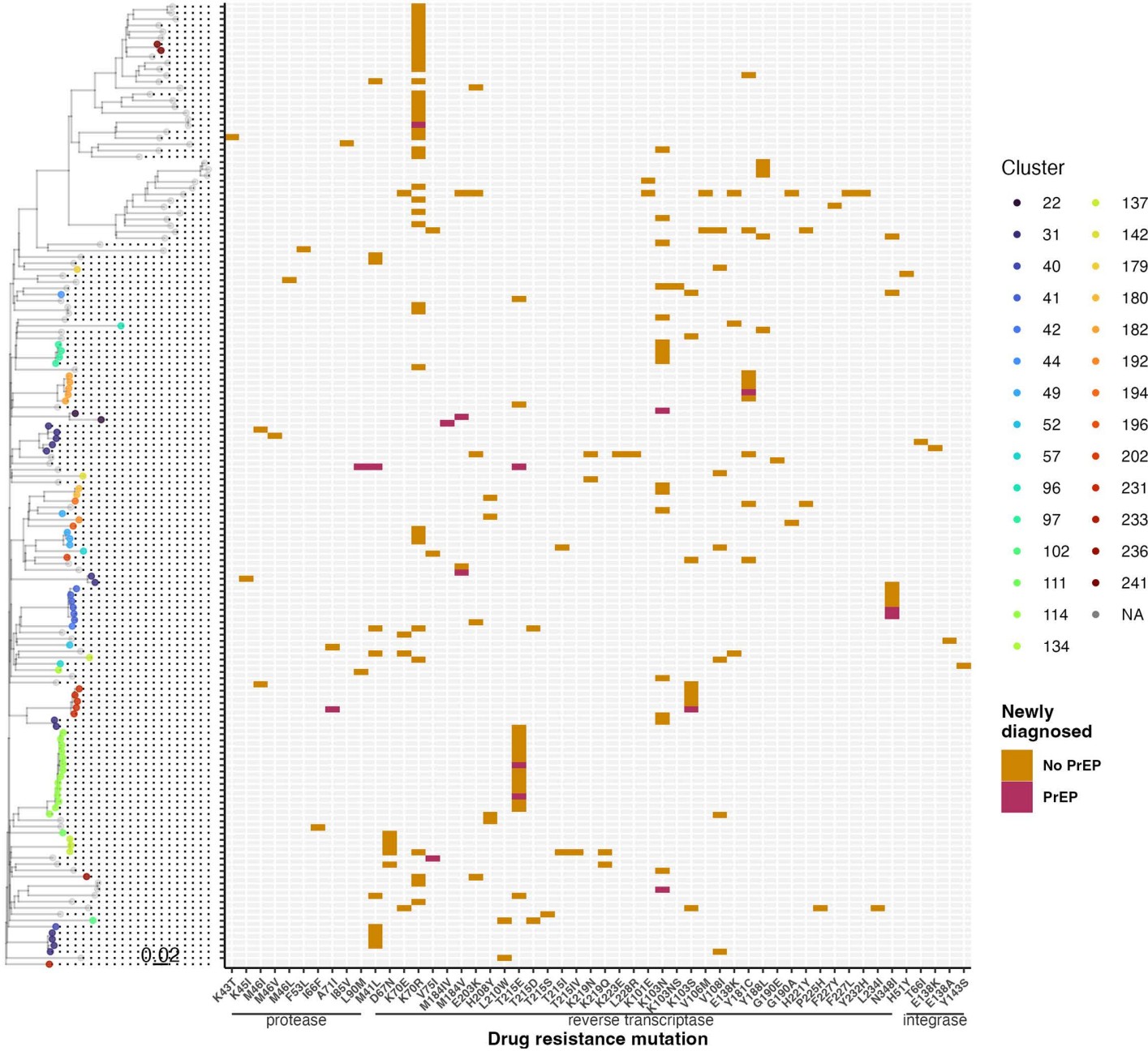

**Fig 2. Baseline drug resistance mutations in newly HIV diagnosed from 2018 to 222 in BC with or without PrEP use.** Restricted to oldest sequences from participants with any baseline resistance-associated mutation (*n* = 154). Mutations are sorted by gene then codon position, and colored by PrEP use. Phylogeny is a bootstrap approximate maximum likelihood tree of partial *pol* sequences trimmed of SDRM codons, rooted on the oldest subtype B sequence, and pruned to newly diagnosed with any baseline drug resistance mutations; scale in substitutions per site. Integrase sequences were not available for all partial *pol* sequences.

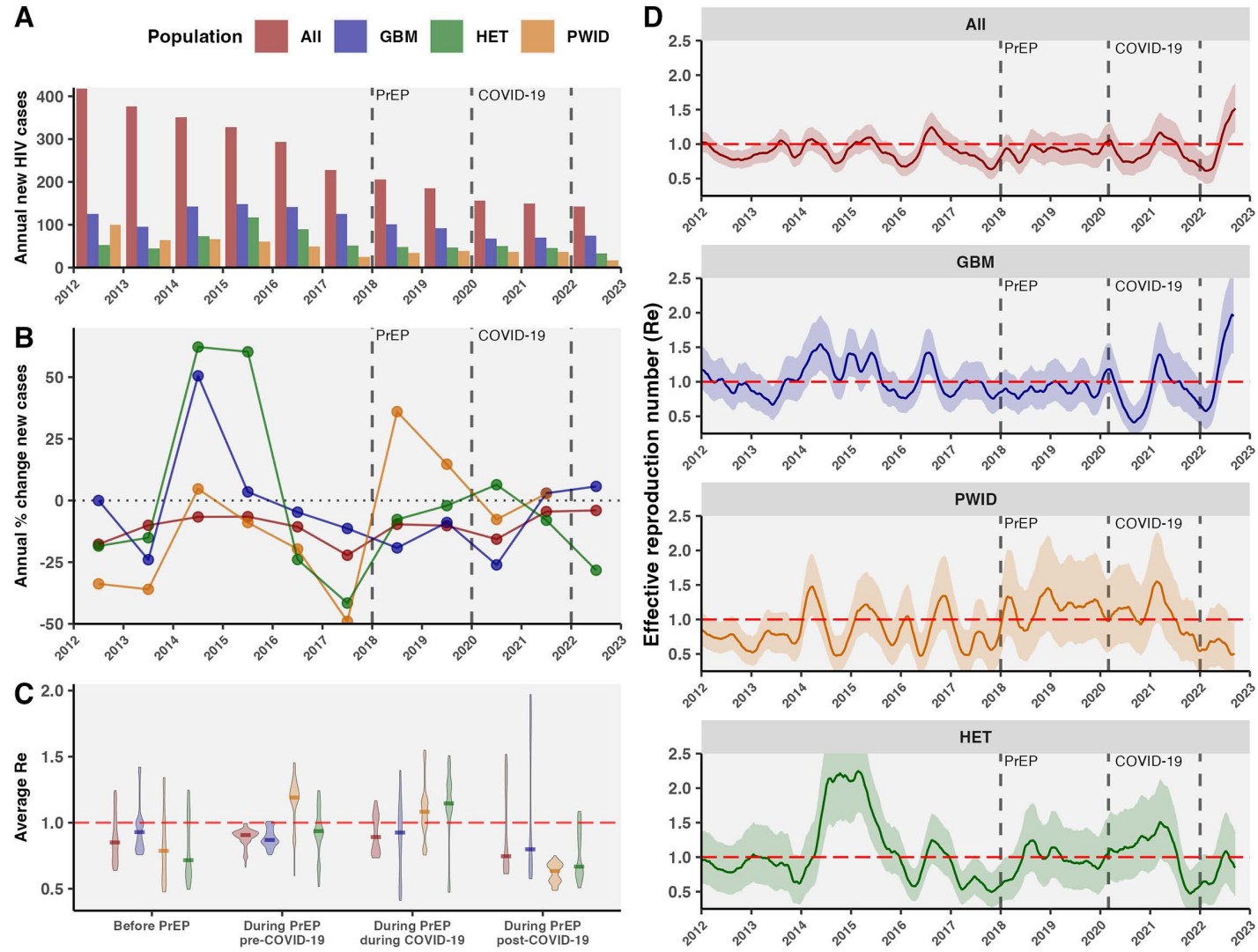

**Fig 3. HIV dynamics in BC from 2015 to 2023 in the context of PrEP and COVID-19. A)** Annual new HIV diagnosed cases (based on first ART date) overall and among key populations: GBM, HET, and PWID. **B)** Annual percentage change in new HIV diagnoses. **C)** Average $R_e$ for periods before PrEP (2016−2017), during PrEP and before COVID-19 (2018-Feb. 2020), during PrEP and during COVID-19 (Mar. 2020−2021), and during PrEP and post-COVID-19 (2022). **D)** Temporal dynamics of $R_e$ overall and in key populations.

volatile $R_e$, peaking in 2017–2018, dropping in 2019, and rising again in late 2019 and 2020. Comparing the PrEP effect on cluster-level $R_e$ in key populations, 50% of GBM clusters had reduced $R_e$ following PrEP, compared to 33.3% of PWID clusters (Fig 4).

## Differential diagnoses averted via PrEP in phylogenetic clusters

Diagnoses averted in phylogenetic clusters via PrEP between 2018 and 2022 were estimated using stochastic branching processes (Fig 5, Table F in S1 Text, Figs AE–AG in S1 Text) [49–51]. Diagnoses averted were calculated as the difference between new diagnoses without and with PrEP simulated with cluster $R_e$ specified as observed $R_e$ (with PrEP) or adjusted $R_e$ (counterfactual without PrEP).

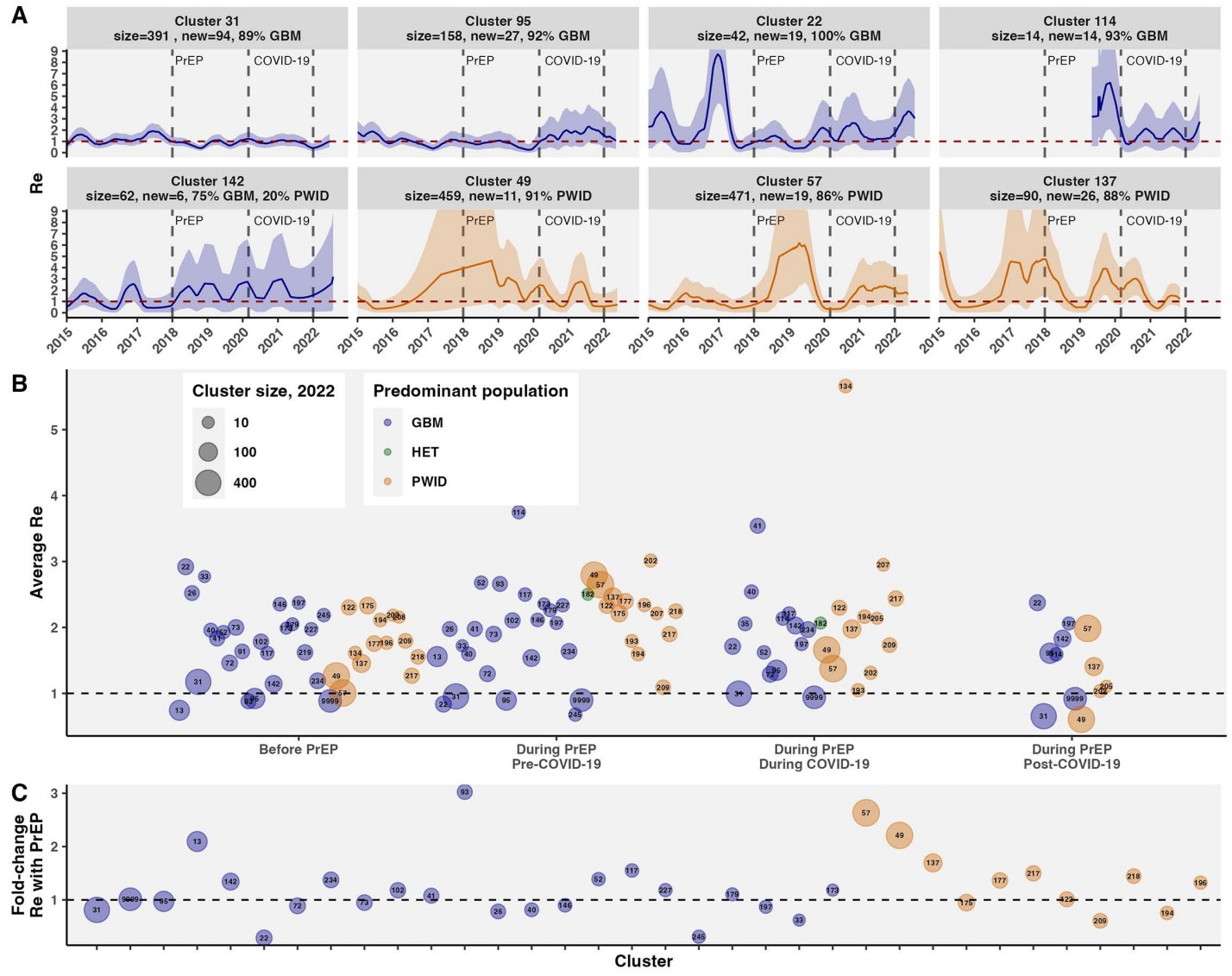

**Fig 4. HIV phylogenetic cluster reproduction number in the context of PrEP and COVID-19. A)** Phylogenetic cluster-level $R_e$ for eight key active clusters in BC from 2015 to 2022. $R_e$ estimated using EpiEstim with gamma-distributed serial interval with mean 1 y and sd 0.5 y, estimating window 0.5 y, and 90 d smoothing. **B)** Piecewise average $R_e$ for active clusters in periods before PrEP, during PrEP and before COVID-19, during PrEP and during COVID-19, and during PrEP and post-COVID-19. Annotated by cluster ID, colored by predominant risk group, and area by cluster size in 2022. **C)** Fold-change (during/before) in active clusters' $R_e$ following PrEP. Clusters sorted on the x-axis by predominant risk group and size in 2022.

PrEP averted the most diagnoses in large and medium-sized GBM-predominant clusters, including clusters 31, 22, 245, and 33 (Fig 6, Fig AG in S1 Text). In cluster 31, we estimated 86.8 new diagnoses (95% CI: 84.2, 89.4) from 2018 to 2022 in simulations based on observed $R_e$ (with PrEP, Fig AE in S1 Text), consistent with the observed 94 new sequenced diagnoses; without PrEP, we estimated up to 115.7 (112.3, 119.1) new diagnoses (Fig AG in S1 Text), amounting to 28.9 (28.2, 9.7) diagnoses averted in cluster 31 in five years. In cluster 22, there were 24.0 (21.7, 26.4) new diagnoses with PrEP (consistent with 19 new diagnoses since 2018), compared to 49.9 (45.9, 54.1) without PrEP, amounting to 25.9 (24.2, 27.7) diagnoses averted. PrEP was also associated with reduced $R_e$ in two medium-sized, relatively young,

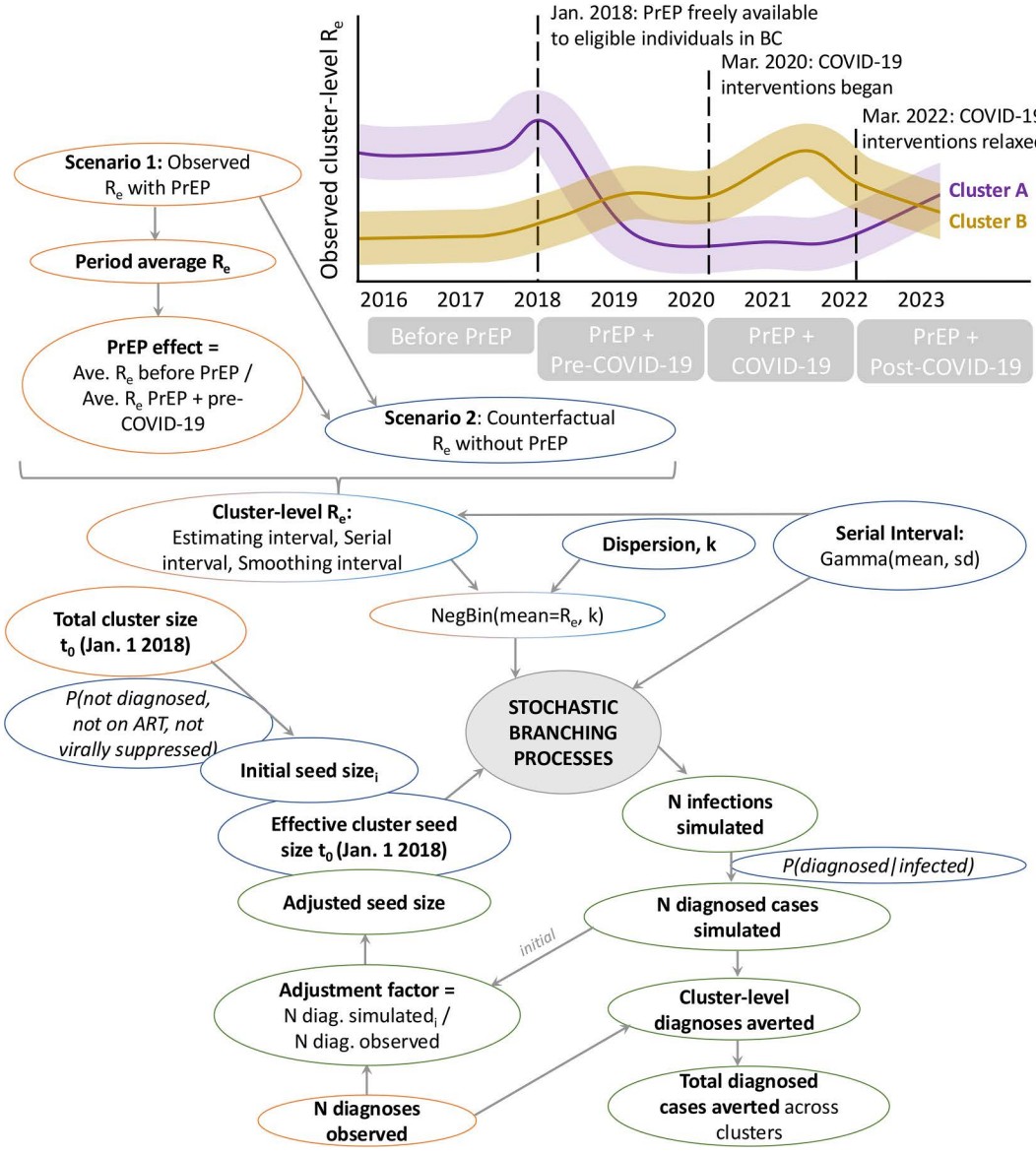

**Fig 5. Flow diagram of stochastic simulations fit to observed scenario with PrEP and for counterfactual scenario without PrEP.** Orange circles were observed parameters, blue were parameters specified based on the literature or PrEP effect, and green were estimated parameters (Table G in S1 Text).

GBM-predominant clusters with one new diagnosis since 2018 (cluster 245, size 2023 = 13, median age = 34; cluster and 33, size 2023 = 10, median age = 37), dropping from $R_e$ well above 1 before PrEP to below 1 during PrEP. We estimated 20.0 (17.9, 22.1) diagnoses averted in cluster 245 and 6.0 (5.2, 64.0) diagnoses averted in cluster 33. In total, among active phylogenetic clusters in BC with HIV diagnoses averted, we estimated 107.6 (99.1, 115.7) diagnoses averted via PrEP in BC between Jan 2018 and Jan 2023, or 21.5 (19.8, 23.1) diagnoses averted per year.

Clusters with negative diagnoses averted represent missed opportunities for PrEP and include PWID-predominant clusters 137, 205, and 49, as well as non-clustered cases (cluster 9,999, predominantly GBM but mixed) and GBM-predominant clusters 142, 234, and 13. Cluster characteristics univariately associated with fewer diagnoses averted

PLOS Medicine

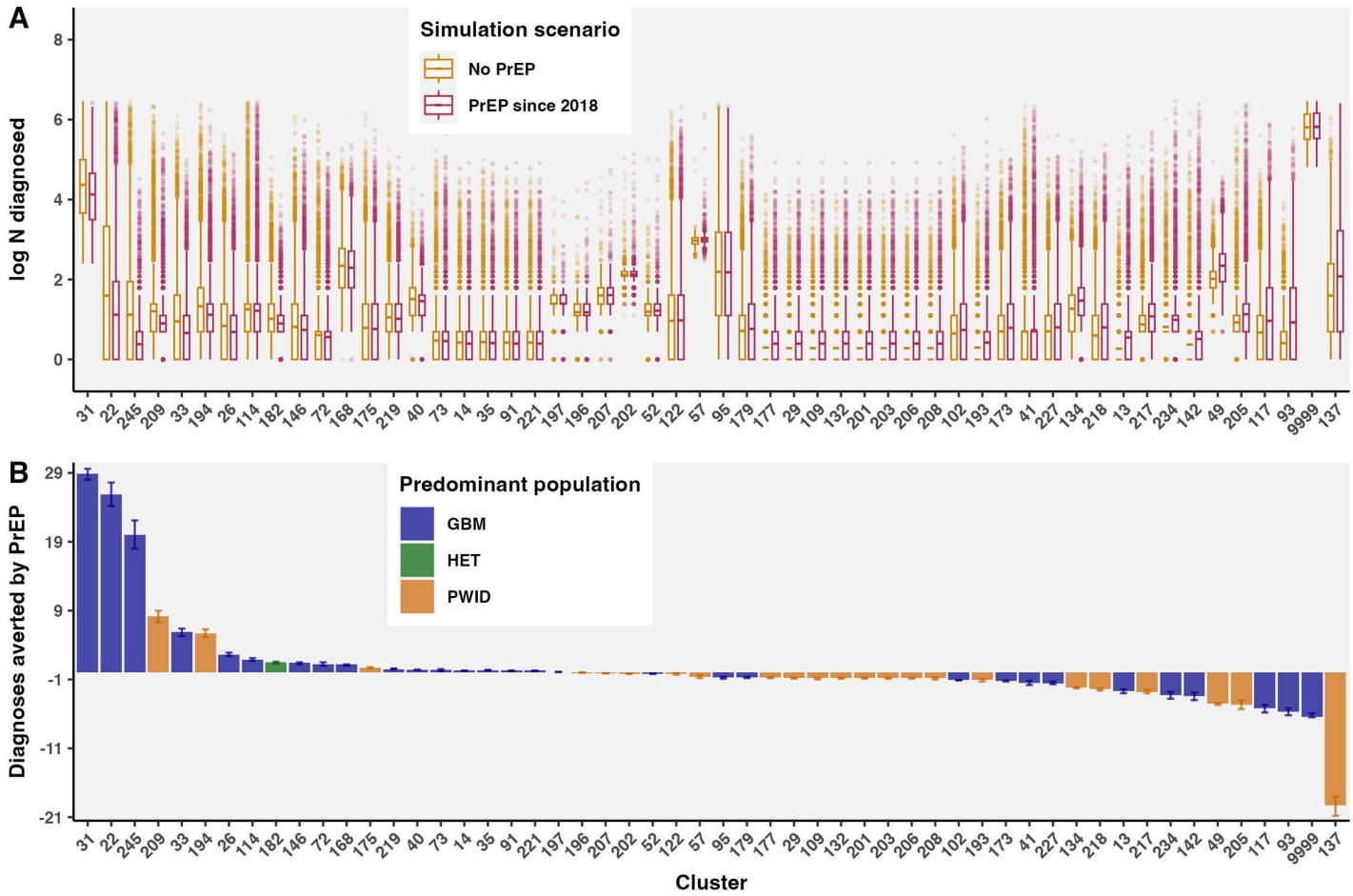

**Fig 6. Diagnoses averted via PrEP in active phylogenetic clusters estimated using stochastic branching processes. A)** Simulations of new diagnoses in clusters from Jan 2018 to Jan 2023 were run with observed cluster $R_e$ (with PrEP since 2018) or with $R_e$ adjusted by the estimated PrEP effect (no PrEP counterfactual scenario). **B)** Diagnoses averted by PrEP from 2018 to 2022 by cluster, estimated as the difference between new diagnoses in simulations with and without PrEP. Bar height represents the mean and error bars represent the bootstrap 95% CI across simulations. Pseudo-cluster 9,999 represents non-clustered cases.

include higher median age, lower proportion of new diagnoses with PrEP use, and living outside of Vancouver Coastal Health Authority. For instance, cluster 142 is comprised of relatively older individuals (median age = 55, median age first ART = 43), with a majority of new diagnoses in Vancouver Island Health Authority (Table E in S1 Text). In a Poisson model of diagnoses averted, clusters with higher median age and lower % GBM had significantly fewer diagnoses averted, adjusted for cluster size at the end of 2017 and % residing in Vancouver Coastal Health Authority (Table H in S1 Text).

## Discussion

Our analyses suggest publicly funded HIV PrEP in BC since 2018, alongside generalized access to free ART under the TasP strategy, was cumulatively associated with more than 100 averted new diagnoses from 2018 to the end of 2022 across active phylogenetic clusters and non-clustered cases, while contributing to baseline NRTI drug resistance in limited cases where PrEP was prescribed or taken inadvertently during acute infection. While PrEP availability was associated with dampened transmission in large GBM-predominant clusters, clusters with fewer cases averted tended to have higher

median age, higher proportion outside of Vancouver Coastal Health, and lower proportion GBM, illuminating prevention gaps and opportunities.

Newly HIV diagnosed PrEP users were more likely than non-PrEP users to be included in phylogenetic clusters and have higher viral diversification rates, which could reflect higher connectivity in the sexual network to some extent, but also systematic differences in risk-taking and health-seeking behavior among those who accessed PrEP. HIV PrEP users may be subject to risk compensation [21], engaging in riskier sex and/or with more partners, but also may get tested more frequently and diagnosed earlier, as supported by their higher baseline median CD4+ count (Table C in S1 Text), both of which could lead to more frequent clustering and higher diversification rates.

Widespread PrEP availability has heterogeneously impacted phylogenetic cluster growth, reflecting underlying differences in cluster contacts' PrEP awareness, eligibility, access, retention, or adherence. We highlighted fewer cases averted via PrEP in clusters with low proportion of previous PrEP use, suggesting that more cases were averted in clusters where new members were more likely to have used PrEP. PrEP uptake, adherence, and retention differ by age, risk perception [55], geography, social norms, substance use, ease of refills, community support, and stigma. Close contacts of members of active phylogenetic clusters can benefit from prioritization for the full spectrum of PrEP education, eligibility, access, and retention. Among HIV-negative GBM not using PrEP in Vancouver in 2018, 28% were uncomfortable asking doctors for PrEP, highlighting the importance of educating physicians as PrEP providers to reduce perceived stigma [55], and exploring potential for alternative care delivery models such as online PrEP services and long-acting formulations.

We found newly HIV diagnosed PrEP users were more likely to have baseline M184I/V conferring reduced susceptibility to NRTIs FTC and 3TC, present in most baseline ART regimens (Table B in S1 Text), and individuals with M184I/V had a short time since last PrEP dispensation including one inadvertent PrEP prescription during acute infection. This suggests PrEP exposure during acute infection may have selected for M184I/V, potentially limiting treatment options containing 3TC or FTC [56]. M184V negatively impacts treatment outcomes for PLWH and increases the likelihood of virological failure and rebound [57]. Baseline M184IV was present among individuals with more recent PrEP dispensations and occurred in an individual with acute HIV infection (false negative) during PrEP initiation. Phylogenetic neighbors of PrEP users with baseline M184V did not carry the mutation, suggesting it was acquired de novo, although it may have been an undetected minority variant. PrEP clients undergo frequent HIV testing, raising the possibility of diagnosis during early acute infection, which could increase the likelihood of detecting M184V. The lack of evidence of onward transmission of M184I/V corroborates its between-host fitness cost. In the absence of drug selection pressure, M184V disappeared rapidly in a UK cohort [58] and in the Swiss HIV epidemic [59]. We detected transmission of other RAMs with clinical implications within clusters, including K70R, which reduces susceptibility to AZT, and K103N, which reduces susceptibility to nevirapine and efavirenz [38]. Detection of T215E TAM revertant mutation in all 14 new cases within one BC HIV cluster could suggest transmission of a variant with improved fitness relative to T215Y/F [60]. Compensatory mutations may impact the fitness conferred or deterred by the RAMs discussed (evaluated in Fig 2), but formal consideration of epistatic effects could improve our ability to decipher clinical and epidemiological outcomes.

There are several limitations to consider in these analyses. First, our statistical comparisons of mutation frequencies between newly diagnosed with PrEP use ($n = 39$) or without ($n = 566$) were imbalanced, reducing the statistical power. Ongoing monitoring of this cohort or pooling of data across jurisdictions would improve the statistical power to detect differences in drug resistant mutation frequencies across groups based on PrEP use patterns. Limitations in the identification of clusters and simulation of their growth include uncertainty in the phylogenetic reconstruction, exacerbated by the absence of undiagnosed individuals and those outside of BC. In this analysis, we focus on transmission between BC residents and assume a densely sampled local epidemic, which is supported by our finding that 77% and 54% of new diagnoses were a part of pre-existing clusters with and without PrEP, respectively. Since clusters are partially sampled, this could lead to underestimating their observed or simulated growth. Sharing anonymized HIV data with metadata across jurisdictions is challenging due to the need to protect patient privacy and confidentiality, therefore there is a need to

develop secure and equitable data-sharing agreements to better understand HIV transmission dynamics on larger scales. Patient confidentiality also dictates that phylogenetic clusters must comprise at least five individuals, which prevents the identification of small clusters that are instead grouped into a pseudo-cluster representing all non-clustered individuals for simulations. The pseudo-cluster represents a heterogeneous population of PLWH, for which some subpopulations may have underestimated cases averted if there was more growth potential in the absence of PrEP than modeled (small GBM clusters), while other subpopulations may have had overestimated cases averted if new cases were similarly low with or without PrEP (virally suppressed recent migrants, for example). Stratifying the non-clustered population into more homogeneous subpopulations could improve model accuracy.

Cluster identification is also sensitive to the pairwise tree or genetic distance threshold, tree inference methods, and use of bootstrap support, leading to differential detection of differences in transmission and sampling [61]. While HIV clusters can be identified using TN93 genetic distance through HIV-TRACE [62], the patristic distance methodology has the advantage of considering recent shared ancestry, with an underlying evolutionary model and bootstrap support. We found phylogenetic clustering identified through patristic distance was concordant with viral lineage-level diversification rate towards evaluating risk factors for elevated transmission risk [27], could be prioritized using the diversification rate [29], and delineated populations most negatively impacted by service disruption during COVID-19 [28]. Regardless of clustering method applied, we illustrate heterogeneity in clusters' transmission dynamics that can be used to more granularly simulate epidemic scenarios.

Our estimations of averted infections are sensitive to clusters' $R_e$, with its underlying uncertainty, smoothing, and serial interval, as well as dispersion and the seed size. Cluster $R_e$ CIs were wide when cases were sparse (Fig 4, Fig X in S1 Text), especially for small and medium-sized clusters. We addressed this by comparing estimation and smoothing intervals for $R_e$ to maximize signal and reduce noise, and screening for wide CIs. Further reducing noise in time-series data could improve $R_e$ estimates for low incidence [63]. Simulations could be impacted by inaccurate $R_e$ due to noise or delayed diagnoses, as well as by inaccurate initial seeds, justifying the calibration of the initial number of infectious individuals for each cluster to yield a similar number of new cases as observed. The specification of observed cluster $R_e$ partially controls for co-occurring epochal effects in simulations. For instance, the COVID-19 pandemic led to reduced access to treatment and prevention resources like safe injection sites, resulting in elevated growth of some PWID-predominant HIV clusters during COVID-19 [28]. The effects of COVID-19 were excluded from estimating the effect of PrEP on cluster transmission by estimating the fold-change of average $R_e$ in the period preceding PrEP to the period with PrEP and before COVID-19 interventions (Jan 2018 to Jan 2020). Stochastic simulations also incorporated uncertainty through the number of secondary infections, as drawn from a negative binomial distribution defined by a mean of $R_e$ and dispersion parameter $k$. It is possible that while accommodating infrequent superspreading, we could have overestimated the number of secondary infections in some cluster simulations; this is why the seed size and hyperparameters were optimized to minimize the difference between observed cases with PrEP compared to simulated in the scenario with PrEP. $R_e$ estimates depend on the serial interval, i.e., the time between successive infections' symptom onset [64], which for HIV can vary widely as transmission could occur during acute infection (45% of transmissions in the first year in a Detroit MSM cohort) [65] or later if an individual has viral rebound (to a detectable VL more than 200 copies/mL) due to treatment interruption or emergence of drug resistance. We compared serial intervals with a mean from 0.5 to five years, reporting one year for the primary analysis under the premise testing and treatment in BC are widely available. Although temporal trends in $R_e$ were robust to serial interval specifications, $R_e$ estimates carry uncertainty forward to estimates of cases averted.

In light of the potential for selection of drug resistance mutation M184I/V and its negative impact on clinical outcomes [57], we recommend reconsidering first-line ART regimens containing TDF or FTC if PrEP use is known or suspected during acute infection. We commend updates to BC PrEP eligibility guidelines to be inclusive of transgender people, and support further inclusivity of PWID and female sex workers. Our cluster simulations suggested more cases could be prevented through improved education, access, and retention for PrEP for older individuals and those residing outside

of Vancouver, for which prescribing physicians play a key role. Future research should consider a comprehensive cost-benefit analysis of widespread PrEP availability, incorporating benefits based on diagnoses averted, the approximate lifetime cost of each HIV diagnosis, estimated as $250,000 Canadian dollars in 2018 [66], as well as costs for drugs, clinic visits, diagnostic tests, and administrative hours [67]. It is unclear the extent to which these results and methodologies are generalizable to other jurisdictions with lower testing rates or greater barriers to accessing HIV PrEP, however, cluster-level epidemiological monitoring and simulations should be feasible in moderate to well-sampled epidemics.

Our findings highlight the success of PrEP towards reducing HIV transmission in BC, alongside interventions such as combination ART [68,69], BC TasP strategy to optimize the control of HIV/AIDS [70,71], and safe injection sites [72]. Ongoing phylogenetic monitoring of clusters provides an epidemiologically relevant delineation of subpopulations to evaluate differences in PrEP and other interventions' effectiveness, investigate transmission of resistance-associated mutations, as well as direct prioritized treatment and prevention resources.

## Supporting information

**S1 Text. Supplementary materials. Table A. HIV-1 antiretroviral therapy (ART) drug names, abbreviation codes, and classes.** Drug classes include nucleoside analogue reverse transcriptase inhibitor (NRTI); non-NRTI (NNRTI); protease inhibitor (PI); and integrase strand transfer inhibitor (INSTI). Excludes broadly neutralizing antibodies, capsid inhibitors, and other classes). Components of TDF/FTC oral PrEP are marked in **bold**. **Table B. First and most recent drug regimens of newly diagnosed PrEP users. Table C. Characteristics of newly diagnosed PrEP users and non-PrEP users from 2018 to 2022.** Proportions compared using chi-squared test and medians compared by Kruskal–Wallis tests. **Table D. Sociodemographic and clinical factors associated with clustering among newly diagnosed PrEP users and non-PrEP users from 2018 to 2022.** P-values reported for two-sided chi-squared tests for categorical variables and Kruskal–Wallis tests for numeric variables. **Fig A. New diagnoses in active HIV phylogenetic clusters in BC since 2018**. **A)** Total new diagnoses in active clusters, annotated with cluster ID; cluster size at the end of study period in December 2022. Bars are colored by PrEP use. **B)** Monthly new diagnoses that clustered during the study period, colored by PrEP use. **C)** New diagnoses in the study period by cluster. Annotated with cluster ID for the eight largest clusters. **Fig B. Clustering and lineage-level viral diversification rate among newly diagnosed with and without PrEP use. A)** Comparison of the proportion of individuals who clustered (membership, yes or no) by PrEP use, using a two-sided chi-squared test. **B)** Distribution of viral lineage-level diversification rate among newly diagnosed PrEP users and non-PrEP users, compared using a Kruskal–Wallis test. **C)** Viral lineage-level diversification rates were compared by PrEP use and clustering using pairwise Mann–Whitney tests. **Fig C. Distribution of phylogenetic cluster sizes identified using multiple rooting strategies.** Trees were rooted using midpoint (Fasttree default), midpoint then binarized, root-to-tip regression, or outgroup rooting on oldest subtype B, G, or H. Annotations for the maximum and mean cluster size, and total number of clusters. **Fig D. Cumulative detection of baseline treatment–selected mutations (TSMs) among newly diagnosed 2018−2022. Fig E. Proportion newly diagnosed with or without PrEP with any baseline TSM or SDRM according to Stanford HIVdb definitions, by drug class. Fig F. NRTI drug resistance scores in newly diagnosed with and without PrEP**. Scores calculated using Stanford algorithm version 9.6 updated March 9, 2024. Drugs include 3TC, ABC, zidovudine (AZT), stavudine (D4T), DDI, FTC, and TDF. P-values were calculated using Kruskal–Wallis tests. **Table E. Characteristics of active phylogenetic clusters with at least 1 new case since Jan 1, 2018 and size 10 or larger. Fig G. Maximum likelihood phylogenetic tree of HIV cluster 22.** Tree was inferred with partial *pol* alignment stripped of drug resistance mutation codons in IQ-TREE with a generalized time reversible model, and rooted using root-to-tip regression. Nodes annotated with ultrafast bootstrap support values. Tips colored for viruses from newly diagnosed (2018−2022) PrEP users and non-PrEP users, and annotated with NRTI resistance-associated mutations including M184IV. Multiple sequences shown per participant. **Fig H. Maximum likelihood phylogenetic tree**

of HIV cluster 31. **Fig I. Maximum likelihood phylogenetic tree of HIV cluster 49. Fig J. Maximum likelihood phylogenetic tree of HIV cluster 57. Fig K. Maximum likelihood phylogenetic tree of HIV cluster 95. Fig L. Maximum likelihood phylogenetic tree of HIV cluster 114. Fig M. Maximum likelihood phylogenetic tree of HIV cluster 137. Fig N. Maximum likelihood phylogenetic tree of HIV cluster 142. Fig O. HIV incidence and prevalence in BC based on DTP participants, in the context of PrEP availability. A)** Annual new HIV cases in BC from 1987 to the end of 2022, estimated by the date of ART, excluding those with previous ART history (HIV-positive migrant). **B)** New HIV cases in BC per 100,000 per year calculated quarterly from 2012 to 2022, normalized to BC population size by quarter [73]. **C)** HIV prevalence per 100,000 in BC from 1987 to 2022, calculated as the cumulative sum of new cases and HIV-positive migrants, subtracting the number of deaths and emigrants. **D)** HIV growth rate per year in BC from 2012 to 2022, calculated as new cases divided by the prevalent population size in each calendar month converted to per year. Dotted vertical lines denote PrEP availability in January 2018 onwards and COVID-19 pandemic interventions from March 2020 to January 2022. **Fig P. Quarterly new HIV cases from January 2012 to February 2023 by key population**. For **A)** all BC residents, **B)** GBM, **C)** PWID, and **D)** HET. Individuals with multiple risk factors are represented in multiple panels. Dotted vertical lines denote PrEP availability in January 2018 onwards and COVID-19 pandemic interventions from March 2020 to January 2022. **Fig Q. Percentage of annual new HIV cases within key populations. A)** GBM, **B)** PWID, **C)** HET. Individuals who reported multiple risk factors are represented in multiple panels and not all individuals reported risk factors. Dotted vertical lines denote PrEP availability in January 2018 onwards and COVID-19 pandemic interventions from March 2020 to January 2022. **Fig R. HIV $R_e$ in BC from 2012 to 2022 under varying serial interval assumptions.** Instantaneous $R_e$ was estimated in EpiEstim with gamma-distributed serial intervals for multiple parameter sets with means (by columns) of 0.5 year (y) (182 days (d)), 1 y (365 d), 2 y (730 d), or 5 y (1,825 d); standard deviations (stdev or sd; by rows) of 0.5 y, 1 y, or 2 y; and estimate windows (by color) of 0.25 y (91 d), 0.5 y, or 1 y. $R_e$ were smoothed with $k = 90$ d, right-aligned (i.e., average over past 90 d). **Fig S. HIV $R_e$ in BC as in Fig R, with no smoothing. Fig T. HIV $R_e$ in BC as in Fig R, with 30 d smoothing. Fig U. HIV $R_e$ in BC as in Fig R, with 365 d smoothing. Fig V. HIV $R_e$ among GBM in BC from 2012 to 2022 under multiple serial interval assumptions.** $R_e$ estimated with gamma distributed serial intervals for multiple parameters with means (by columns) of 0.5 y (182 d), 1 y (365 d), 2 y (730 d), or 5 y (1,825 d); standard deviations (by rows) of 0.5 y, 1 y, or 2 y; and estimate intervals (by color) of 0.25 y (91 d), 0.5 y, or 1 y. $R_e$ were smoothed with $k = 90$ d, right-aligned. **Fig W. HIV $R_e$ among PWID in BC from 2012 to 2022 under multiple serial interval assumptions.** $R_e$ was estimated with gamma distributed serial intervals for multiple parameters with means (by columns) of 0.5 y (182 d), 1 y (365 d), 2 y (730 d), or 5 y (1,825 d); standard deviations (by rows) of 0.5 y, 1 y, or 2 y; and estimate intervals (by color) of 0.25 y (91 d), 0.5 y, or 1 y. $R_e$ were smoothed by 90 d, right-aligned. **Fig X. HIV $R_e$ across large (size in 2022 > 9) and active (new cases since 2018 > 2) clusters, estimated with gamma-distributed serial interval mean 1 y, standard deviation (sd) 0.5 y, estimating window 0.5 y, and 90 d smoothing.** Excludes values where the confidence interval (CI) width exceeded 10 due to sparse cases. Cluster 9,999 is all non-clustered cases. Color denotes predominant population (blue: GBM, orange: PWID, and green: HET). **Fig Y. HIV $R_e$ across large and active clusters, estimated with gamma-distributed serial interval _mean 2 y_, sd 0.5 y, estimating window 0.5 y, and 90 d smoothing. Fig Z. HIV $R_e$ across large and active clusters, estimated with gamma-distributed serial interval mean 1 y, _sd 1 y_, estimating window 0.5 y, and 90 d smoothing. Fig AA. HIV $R_e$ across large and active clusters, estimated with gamma-distributed serial interval mean 1 y, sd 0.5 y, _estimating window 1 y_, and 90 d smoothing. Fig AB. Quarterly new cases in large active clusters by PrEP use.** Cluster 9,999 represents non-clustered cases. **Fig AC. Epochal PrEP effect and COVID-19 effect on clusters' $R_e$.** Epochal effects were calculated as fold-change in the piecewise average $R_e$ in the PrEP period, compared to before PrEP (during PrEP/pre-COVID, Jan 2018–Feb 2020 vs. before PrEP, Jan 2016–Dec 2017), COVID-19 (during COVID, Mar 2020–Dec 2021 vs. during PrEP/pre-COVID), and post-COVID-19 (Jan–Dec 2022 vs. during COVID). Point size represents cluster size in 2022, color is predominant population. Restricted to medium (≥10 size), active (≥1 case since 2018) clusters. Box plots show median and

interquartile range of predominant groups' epochal effects. **Table F. Calibration of simulation cluster seed size (number of infectious individuals at time zero).** Active clusters shown. Cluster 9,999 represents non-clustered individuals. **Fig AD. Observed and adjusted (in the absence of PrEP) cluster $R_e$.** Includes clusters with at least one new case since 2018 and size of at least 10 in January 2023. **Fig AE. Stochastic branching processes recapitulate observed cluster growth.** Observed number of new samples 'PrEP (obs.)' in large, active clusters from 2018 to 2022 compared to distribution of new samples across 2000 simulations using clusters' observed $R_e$ 'PrEP (sim.)'. **Fig AF. Simulations of cluster samples in the absence of PrEP.** Observed number of new samples (among new diagnoses) 'PrEP (obs.)' in large, active clusters from 2018 to 2022 compared to distribution of new samples across 4,000 simulations using clusters $R_e$ adjusted by PrEP effect 'no PrEP (sim.)'. **Fig AG. Simulations of cluster diagnoses in the absence of PrEP.** Observed number of new diagnoses 'PrEP (obs.)' in large, active clusters from 2018 to 2022 compared to distribution of new diagnoses across 4,000 simulations using clusters $R_e$ adjusted by PrEP effect 'no PrEP (sim.)'. **Table G. Parameters of stochastic branching process simulations were specified based on the literature, observed or estimated at the cluster-level.** Parameters were grouped into types by whether they were specified based on literature, observed (trends) at cluster-level, estimated at cluster-level in simulations. Standard deviation (sd). Uncertainty in $R_e$ estimation was explored for variables marked **. **Table H. Poisson model of diagnoses averted across clusters**. Counts were normalized to be positive integers (minimum averted added to all). Exponentiated coefficients reported as mean with lower and upper 95% CIs. Significant adjusted relationships in bold.
(PDF)

**S2 STROBE checklist. STROBE Statement—checklist of items that should be included in reports of observational studies.** Checklist available from https://www.strobe-statement.org/checklists/.
(PDF)

**S3 File. Ethics protocol, application, and certificates.**
(PDF)

## Acknowledgments

We thank DTP participants and staff, as well as colleagues in the BC-CfE Laboratory.

## Author contributions

**Conceptualization:** Angela McLaughlin, Jeffrey B. Joy.

**Data curation:** Angela McLaughlin, Junine Toy, Paul Sereda, Jason Trigg, Chanson J. Brumme.

**Formal analysis:** Angela McLaughlin.

**Funding acquisition:** Jeffrey B. Joy.

**Methodology:** Angela McLaughlin, Junine Toy, Chanson J. Brumme, Jeffrey B. Joy.

**Project administration:** Jeffrey B. Joy.

**Resources:** Jeffrey B. Joy.

**Supervision:** Jeffrey B. Joy.

**Validation:** Angela McLaughlin, Jeffrey B. Joy.

**Visualization:** Angela McLaughlin.

**Writing – original draft:** Angela McLaughlin.

**Writing – review & editing:** Junine Toy, Vincent Montoya, Paul Sereda, Jason Trigg, Mark Hull, Chanson J. Brumme, Rolando Barrios, Julio S. G. Montaner, Jeffrey B. Joy.

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
