## [Editor Report · Decision Letter 0]

2 May 2025

Dear Dr Joy,

Thank you for submitting your manuscript entitled "Impact of pre-exposure prophylaxis on HIV drug resistance and phylogenetic cluster growth" for consideration by PLOS Medicine.

Your manuscript has now been evaluated by the PLOS Medicine editorial staff as well as by an academic editor with relevant expertise and I am writing to let you know that we would like to send your submission out for external peer review. My apologies for accidentally triggering the earlier rejection letter.

In discussing the manuscript with an academic editor, a few points were raised that you might consider addressing at this point to avoid their being raised in external review:

The academic editor felt that more clarity would be helpful as you perform many different analyses that seem to draw from different populations, some based on real data, and some simulated. Please ensure that the populations and data sources for the analyses are clearly identified at each point in order for the reader to better understand how the insights build on top of each other. What is modeled in each instance needs to be clearly stated.

Please also note that the academic editor felt that the writing in general requires greater clarity so that how the analyses fit together is more transparently articulated.

Please also note that before we can send your manuscript to reviewers, we need you to complete your submission by providing the metadata that is required for full assessment. To this end, please login to Editorial Manager where you will find the paper in the 'Submissions Needing Revisions' folder on your homepage. Please click 'Revise Submission' from the Action Links and complete all additional questions in the submission questionnaire. Please deposit any new code generated for these analyses in a public repository.

Please re-submit your manuscript within two working days, i.e. by May 06 2025 11:59PM.

Kind regards,

Alison Farrell, Ph.D.

Senior Editor

PLOS Medicine

---

## [Decision Letter · Decision Letter 1]

7 Aug 2025

Dear Dr Joy,

Many thanks for submitting your manuscript "Impact of pre-exposure prophylaxis on HIV drug resistance and phylogenetic cluster growth" (PMEDICINE-D-25-01591R1) to PLOS Medicine. The paper has been reviewed by subject experts and a statistician; their comments are included below and can also be accessed here: ********

As you will see, the reviewers find the study interesting, but raise important points that must be addressed in a revised manuscript. In particular, reviewer 1 requires additional analyses, reviewers 2 and 3 require clarification, justification and support for your definition of transmission clusters, and reviewer 3 is concerned that the conclusions may not be generalizable in view of the low number of HIV infections among PrEP users. The editors require that you revise the presentation to better highlight the novel insights derived from the study and how they might guide public or clinical health interventions. Please also ensure that the language in the manuscript adheres to PLOS guidelines and is person-centered and non-stigmatizing. After discussing the paper with the editorial team and an academic editor with relevant expertise, I'm pleased to invite you to revise the paper in response to the reviewers' comments. We plan to send the revised paper to some or all of the original reviewers, and we cannot provide any guarantees at this stage regarding publication. We will also seek the input of an independent statistician on a revised manuscript.

Please further see the requested edits in the attached file.

We ask that you submit your revision by Aug 28 2025 11:59PM. However, if this deadline is not feasible, please contact me by email, and we can discuss a suitable alternative.

Don't hesitate to contact me directly with any questions (afarrell@plos.org).

Best regards,

Alison

Alison Farrell, Ph.D.

Senior Editor

PLOS Medicine

afarrell@plos.org

Comments from the reviewers:

Reviewer #1: The study examines the effect of PrEP introduction on the individual-level (i.e. emergence of drug resistance) and on population-level HIV transmission dynamics. Using HIV molecular epidemiology, the authors show that newly-diagnosed PrEP users were more likely to join existing clusters and were more likely to have M184V DRM. Using cluster-level Re inference, the authors estimated the number of new HIV infections averted due to PREP and also present RE changes over time in clusters comprising various key populations living with HIV. The authors innovatively use molecular epi to show the differential effects of PrEP on new infections in different key populations. Overall, this paper will make a strong contribution to public health knowledge, I only have a few comments to help improve it.

My main concerns are:

1) Some of the observed elevated clustering is expected in higher-risk individuals that would be eligible for PrEP. Instead of comparing characteristics in newly-diagnosed PrEP users to all non PrEP users, a comparison to those non PrEP users who were eligible for PrEP could be more fitting.

2) The number of averted infections in clusters was simulated using the cluster definition of 5+ infections and the observed cluster distribution. I suggest some sensitivity analysis on cluster size distribution for this analysis, or at least specify/discuss that the observed is likely the lower bound of the number of averted infections.

Some other minor comments:

1. Line 108 - please clarify whether the hypothesis was focused on any clustering, or clustering within the PrEP users networks.

2. Cost analysis seems redundant as the authors mention that it does not take the alternative costs (i.e. cost of Prep) into account. As is seems like an unnecessary addition to the paper.

3. Higher diversification rates are also expected in PrEP users since they are more likely to be diagnosed during acute infection.

4. Would be helpful to mention whether nearest phylogenetic neighbours to non-PrEP users with M184I/V had M184I/V (to support that it's likely direct transmissions and we're not missing intermediates, which would help the point made for PrEP users)

5. Integrase DRM are in the table but not discussed in the text

6. Figure 1B would be more helpful with resolved polytomies

7. Did you use any threshold to define clusters as "predominantly" GBM?

8. Line 427 - does Prep access here mean Prep use or Prep eligibility?

Reviewer #2: From the methods it is not clear how authors defined clusters, as they mentioned that they used cutoff and bootstrap as indicated in PMC4853759, but not further steps from those methods to ensure proper cluster identification. Can authors use a phylogenetic cluster informed method like PhyloPart or genetic distance based HIV-trace to define clusters and compare with their method? It would be interesting to know how diverse PREP-users sequences are from non-PREP. Since authors used longitudinal data per patient (sometimes up to 35 sequences per patient), are clusters comprised of mainly patient longitudinal data? How large were clusters containing more homogenous clusters (longitudinal patient data) compared to heterogenous clusters (diverse patients, no longitudinal data).

Another question about cluster definition, is whether the clusters are BC unique or are intermixed with sequences from other Canadian provinces or foreign countries. Can authors add GenBank sequences to check the monophyly of their clusters?

What are the epidemiological, clinical and resistance profile of the 9 PREP sequences that non clustered? are these from a unique patient or 9 patients?

Based on the results that PrEP users were more likely to have baseline M184I/V, what authors would suggest as improved/new guidelines to public health officials about the issue?

Reviewer #3: In this paper the authors compare newly diagnosed PrEP users (n=39) in British Columbia to newly diagnosed non-PrEP users (n=566) over the period of 2018-2022. Transmission clusters were identified by phylogenetic analysis and their composition analyzed to assess the efficacy of PrEP. The findings show that newly HIV diagnosed PrEP users were significantly more likely than non-PrEP users to join clusters and carry baseline nucleoside-analogue reverse transcriptase inhibitor (NRTI) resistance mutation M184I/V. The authors also show through simulation that PrEP averted approximately 20 new HIV diagnoses per year in British Columbia since 2018, and infrequently contributed to baseline drug resistance. Overall, the paper is well written and the findings provide an assessment of the efficacy of PrEP. However, the result is not novel, since many studies have shown that PrEP can effectively reduce the number of transmissions. Additionally the study includes a relative low number of diagnosed PrEP users (n=39), which casts doubts on the robustness of the statistical results. From a technical standpoint, transmission clusters were identified as those clades in the phylogeny including at least five individuals with viruses sharing pairwise patristic distance less than 0.02 substitutions per site in at least 90% of bootstraps support. Although the authors cite Poon et al. 2016 for such a definition, the choice of a 0.02 threshold appears to be arbitrary and unjustified. How would the conclusions change if different threshold had been used?

Any attachments provided with reviews can be seen via the following link: ********

---

* Please upload any figures associated with your paper as individual TIF or EPS files with 300dpi resolution at resubmission; please read our figure guidelines for more information on our requirements: http://journals.plos.org/plosmedicine/s/figures. While revising your submission, please upload your figure files to the PACE digital diagnostic tool, https://pacev2.apexcovantage.com/. PACE helps ensure that figures meet PLOS requirements. To use PACE, you must first register as a user. Then, login and navigate to the UPLOAD tab, where you will find detailed instructions on how to use the tool. If you encounter any issues or have any questions when using PACE, please email us at PLOSMedicine@plos.org.

* CONFIRM FINANCIAL DISCLOSURES, COI, DAS, AND ETHICS STATEMENTS ARE CORRECT.

* Please ensure that the study is reported according to the appropriate guideline and include the completed checklist as Supporting Information. When completing the checklist, please use section and paragraph numbers, rather than page numbers. Please add the following statement, or similar, to the Methods: "This study is reported as per [XXXX] guideline (S1 Checklist)."

FIGURES AND TABLES

SUPPLEMENTARY MATERIAL

REFERENCES

OBSERVATIONAL STUDIES

* Abstract: Please include the study design, population and setting, number of participants, years during which the study took place (enrollment and follow up), length of follow up, and main outcome measures.

* Please ensure that the study is reported according to the STROBE (or appropriate STOBE extension) guideline (available from: https://www.equator-network.org/reporting-guidelines/strobe) and include the completed STROBE (or STROBE extension) checklist as Supporting Information. Please add the following statement, or similar, to the Methods: "This study is reported as per the Strengthening the Reporting of Observational Studies in Epidemiology (STROBE) guideline (S1 Checklist)." When completing the checklist, please use section and paragraph numbers, rather than page numbers.

* [FOR POPULATION HEALTH/REGISTRY STUDIES] Please ensure that the study is reported according to the RECORD guideline (available from https://www.record-statement.org) and include the completed checklist as Supporting Information. Please add the following statement, or similar, to the Methods: "This study is reported as per the Reporting of Studies Conducted using Observational Routinely-Collected Data (RECORD) guideline (S1 Checklist)." When completing the checklist, please use section and paragraph numbers, rather than page numbers.

* [FOR POPULATION HEALTH ESTIMATES] Please ensure that the study is reported according to the GATHER statement (available from https://www.equator-network.org/reporting-guidelines/gather-statement) and include the completed checklist as Supporting Information. Please add the following statement, or similar, to the Methods: "This study is reported as per the Guidelines for Accurate and Transparent Health Estimates Reporting (GATHER) statement (S1 Checklist)." When completing the checklist, please use section and paragraph numbers, rather than page numbers.

* For all observational studies, in the manuscript text, please indicate: (1) the specific hypotheses you intended to test, (2) the analytical methods by which you planned to test them, (3) the analyses you actually performed, and (4) when reported analyses differ from those that were planned, transparent explanations for differences that affect the reliability of the study's results. If a reported analysis was performed based on an interesting but unanticipated pattern in the data, please be clear that the analysis was data driven.

* Please state in the Methods section whether the study had a prospective protocol or analysis plan. If a prospective analysis plan (from your funding proposal, IRB or other ethics committee submission, study protocol, or other planning document written before analyzing the data) was used in designing the study, please include the relevant document(s) with your revised manuscript as a Supporting Information file to be published alongside your study and cite it in the Methods section. A legend for this file should be included at the end of your manuscript. If no such document exists, please make sure that the Methods section transparently describes when analyses were planned, and when/why any data-driven changes to analyses took place. Changes in the analysis, including those made in response to peer review comments, should be identified as such in the Methods section of the paper, with rationale.

MODELLING STUDIES

The following list is derived from Geoffrey P Garnett, Simon Cousens, Timothy B Hallett, Richard Steketee, Neff Walker. Mathematical models in the evaluation of health programmes. (2011) Lancet DOI:10.1016/S0140-6736(10)61505-X:

* If pertinent, please provide a diagram that shows the model structure, including how the natural history of the disease is represented, the process and determinants of disease acquisition, and how the putative intervention could affect the system.

* Please provide a complete list of model parameters, including clear and precise descriptions of the meaning of each parameter, together with the values or ranges for each, with justification or the primary source cited and important caveats about the use of these values noted.

* Please provide a clear statement about how the model was fitted to the data, including goodness-of-fit measure, the numerical algorithm used, which parameter varied, constraints imposed on parameter values, and starting conditions.

* For uncertainty analyses, please state the sources of uncertainties quantified and not quantified [can include parameter, data, and model structure].

* Please provide sensitivity analyses to identify which parameter values are most important in the model. Uncertainty estimates seek to derive a range of credible results on the basis of an exploration of the range of reasonable parameter values. The choice of method should be presented and justified.

* Please discuss the scientific rationale for the choice of model structure and identify points where this choice could influence conclusions drawn. Please also describe the strength of the scientific basis underlying the key model assumptions.

* For studies that develop a prediction model or evaluate its performance, please ensure that the study is reported according to the TRIPOD statement (https://www.equator-network.org/reporting-guidelines/tripod-statement) and include the completed checklist as Supporting Information. Please add the following statement, or similar, to the Methods: "This study is reported as per the Transparent Reporting of a Multivariable Prediction Model for Individual Prognosis Or Diagnosis (TRIPOD) statement (S1 Checklist)." For studies using machine learning, please use the TRIPOD-AI checklist. When completing the checklist, please use section and paragraph numbers, rather than page numbers.

---

## [Decision Letter · Decision Letter 2]

17 Oct 2025

Dear Dr. Joy,

Thank you very much for re-submitting your manuscript "Heterogeneous impacts of HIV pre-exposure prophylaxis on drug resistance and phylogenetic cluster transmission dynamics in British Columbia, Canada" (PMEDICINE-D-25-01591R2) for review by PLOS Medicine.

I have discussed the paper with my colleagues and it was also seen again by three reviewers. I am pleased to say that provided the remaining editorial and production issues are dealt with we are planning to accept the paper for publication in the journal.

[LINK]

We look forward to receiving the revised manuscript by Oct 21 2025 11:59PM.   

Sincerely,

Alison Farrell, Ph.D.

Senior Editor 

PLOS Medicine

plosmedicine.org

Requests from Editors:

** Please confirm that your title complies with PLOS Medicine's style. Your title must be nondeclarative and not a question. It should begin with main concept if possible. "Effect of" should be used only if causality can be inferred, i.e., for an RCT. Please place the study design ("A randomized controlled trial," "A retrospective study," "A modelling study," etc.) in the subtitle (ie, after a colon).

* In the abstract, please include the important dependent variables that are adjusted for in the analyses.

* Please ensure that the Introduction ends with a clear description of the study question or hypothesis.

* Please ensure that all abbreviations are defined at first use throughout the text.

* Please confirm that all numbers presented in the abstract are present and identical to numbers presented in the main manuscript text.

* PLOS defines the “minimal data set” to consist of the data set used to reach the conclusions drawn in the manuscript with related metadata and methods, and any additional data required to replicate the reported study findings in their entirety. Authors do not need to submit their entire data set, or the raw data collected during an investigation. Please submit the following data:

The values behind the means, standard deviations and other measures reported;

The values used to build graphs;

The points extracted from images for analysis.

* For studies in which a novel model is central to the manuscript's findings, authors are responsible for providing the source code needed to replicate the study's findings in a repository (such as GitHub, SourceForge or Bitbucket) or a cloud computing service (such as Code Ocean). Protection of authors’ intellectual property will not be cause for exception. Please explain in the manuscript’s Data Availability Statement how readers can access the shared code.

* The current explanation for lack of code inclusion is not sufficiently justified.

* Please review your manuscript and edit to ensure compliance with our inclusive language requirements https://journals.plos.org/plosmedicine/s/human-subjects-research#loc-categorization

* Please consider avoiding the use of red and green in order to make your figure more accessible (e.g Fig. 3).

* Where data points are discrete, please ensure that they are depicted in the figures as discrete data and not as a continuous line.

** The funding statement should include: specific grant numbers, initials of authors who received each award, URLs to sponsors’ websites. Also, please state whether any sponsors or funders (other than the named authors) played any role in study design, data collection and analysis, the decision to publish, or preparation of the manuscript. If they had no role in the research, include this sentence: “The funders had no role in study design, data collection and analysis, decision to publish, or preparation of the manuscript.”

*Please provide the missing URLs.

** All authors must declare their relevant competing interests per the PLOS policy, which can be seen here: https://journals.plos.org/plosmedicine/s/competing-interests For authors with ties to industry, please indicate whether any of the interests has a financial stake in the results of the current study.

Please include a completed checklist that is most appropriate for your study (e.g. STROBE, GATHER, RECORD):

* For observational studies, please ensure that the study is reported according to the STROBE guideline, and include the completed STROBE checklist as Supporting Information. Please add the following statement, or similar, to the Methods: "This study is reported as per the Strengthening the Reporting of Observational Studies in Epidemiology (STROBE) guideline (S1 Checklist)."

* Did your study have a prospective protocol or analysis plan? Please state this (either way) early in the Methods section.

* Your study is observational and therefore causality cannot be inferred. Please remove language that implies causality and refer to associations instead.

* For all observational studies, in the manuscript text, please indicate: (1) the specific hypotheses you intended to test, (2) the analytical methods by which you planned to test them, (3) the analyses you actually performed, and (4) when reported analyses differ from those that were planned, transparent explanations for differences that affect the reliability of the study's results. If a reported analysis was performed based on an interesting but unanticipated pattern in the data, please be clear that the analysis was data-driven.

*For a population health/registry study, please report that data according to the RECORD guideline (available from https://www.record-statement.org) and include the completed checklist as Supporting Information. Please add the following statement, or similar, to the Methods: "This study is reported as per the Reporting of Studies Conducted using Observational Routinely-Collected Data (RECORD) guideline (S1 Checklist)." When completing the checklist, please use section and paragraph numbers, rather than page numbers

* For population-level health estimates, we as that you report your data according to GATHER and enclose a completed GATHER checklist as a supplementary document. See http://gather-statement.org/ In the checklist please include sufficient text excerpted from the manuscript to explain how you accomplished all applicable items.

For the simulations of cluster growth, please consider whether these guidelines for modeling should be adopted (derived from Geoffrey P Garnett, Simon Cousens, Timothy B Hallett, Richard Steketee, Neff Walker. Mathematical models in the evaluation of health programmes. (2011) Lancet DOI:10.1016/S0140-6736(10)61505-X):

* Please provide a diagram that shows the model structure, including how the disease natural history is represented, the process and determinants of disease acquisition, and how the putative intervention could affect the system.

* Please provide a complete list of model parameters, including clear and precise descriptions of the meaning of each parameter, together with the values or ranges for each, with justification or the primary source cited, and important caveats about the use of these values noted.

* Please provide a clear statement about how the model was fitted to the data including where relevant goodness-of-fit measure, the numerical algorithm used, which parameter varied, constraints imposed on parameter values, and starting conditions.

* For uncertainty analyses, please state the sources of uncertainties quantified and not quantified this can include parameter, data, and model structure.

* Please provide sensitivity analyses to identify which parameter values are most important in the model. Uncertainty estimates seek to derive a range of credible results on the basis of an exploration of the range of reasonable parameter values. The choice of method should be presented and justified.

* Please discuss the scientific rationale for this choice of model structure and identify points where this choice could influence conclusions drawn. Please also describe the strength of the scientific basis underlying the key model assumptions.

Text edits:

Use the active voice throughout.

Line 18: replace ‘taken’ with ‘used’

Line 23: add ‘against HIV acquisition after ‘effectiveness’

Please revise and shorten sentence line 24-28. ‘including heterogeneity’ is not clear. ‘propose key groups..’ seems adjacent to the main findings.

In the Methods and Findings: please state the size and source of the cohort and clarify that you are using sequence data to identify phylogenetic clusters and baseline resistance mutations.

Line 31: please qualify that newly diagnosed is with HIV.

Lines 29-41 can be slightly abbreviated, but specific results need to be added. For example, revise to: “Using a retrospective cohort design, we evaluated the frequencies of baseline drug resistance mutations and membership in phylogenetic clusters among newly diagnosed PrEP users in BC (n=39) compared to non-PrEP users (n=566) diagnosed from 2018 to 2022. Newly HIV-diagnosed persons using PrEP were significantly more likely [add 95% CI] than newly diagnosed non-PrEP users to have/carry viruses that formed part of phylogenetic clusters and contained the baseline nucleoside analogue reverse transcriptase inhibitor (NRTI) resistance mutation M184I/V”.

*Can PrEP ‘users’ be revised to more inclusive language?

*Please don’t add conclusions to the Methods and Findings section of the Abstract, and remove causal language. E.g. “Most diagnoses were averted in large…” As this is a modeling study, causal language should be removed and associations described.

Lines 42-43: this clause “and fewer diagnoses were averted in clusters with a higher median age, lower proportion of new diagnoses with PrEP use” is not clear to the general reader.

Please add limitations of the methodology as the last sentence of the Methods and Findings section.

Line 50: “and 50 pinpoint groups for prioritized PrEP services.” Please temper ‘pinpoint’ Please also note that the heterogeneity mentioned is not clear in the Methods and Findings subsection.

Lines 74-78: please break in 2 for clarity. However please note that all of the results do not need to be reiterated in the Author Summary.

Please also note that the Conclusions section does not explain what the phylogenetic clustering might mean. Overall the Abstract needs to better communicate what the key findings are in this report and what it adds to the literature.

Line 122: THE is capitalized. Please correct.

Line 151 and elsewhere: suggest to delete ‘join’ and replace with ‘form part of’, given that ‘join’ is active and seems inappropriate in this instance.

For clarity: is preventing 20 new HIV diagnoses a per cluster estimate? If so, please rephrase.

Line 345: please rephrase ‘a clinical miss’

For all confidence intervals, please use a comma rather than a hyphen.

Please ensure that no new results are reported in the Discussion and that all findings are discussed in the Results section.

Please ensure that the power calculation is discussed in the Methods.

Line 493: where does the number ‘100’ come from? Here it would be helpful to understand the

% of projected at risk persons accessing PrEP.

The Discussion exceeds 200 words. Please remove repetition of results and streamline. Focus on the novel findings that might inform other studies and ensure the Discussion is interpretable by a general reader. Please do not repeat information in the Introduction and limit recapitulation of the results. Please communicate what is of most interest to public health efforts and interventions.

Comments from Reviewers:

Reviewer #1: The authors answered all of my questions and satisfactory addressed my concerns. Where additional analyses were not possible, I understand the justifications.

Reviewer #2: the authors have satisfactorily addressed my comments, I do not have further revisions to suggest.

Reviewer #3: I must compliment the authors for the excellent job done in revising the manuscript. All concerns from the reviewers were, in my opinion, satisfactorily addressed. In particular, I commend the authors for clarifying and discussing the potential limitations of their work. This is now an original and well written manuscript.

[LINK]

---

## [Editor Report · Decision Letter 3]

12 Nov 2025

Dear Dr Joy, 

On behalf of my colleagues and the Academic Editor, Aaloke Mody, I am pleased to inform you that we have agreed to publish your manuscript " 

Heterogeneous impacts of HIV pre-exposure prophylaxis (PrEP) on drug resistance and phylogenetic cluster transmission dynamics in British Columbia, Canada: a retrospective cohort and simulation study" (PMEDICINE-D-25-01591R3) in PLOS Medicine.

PRESS

Sincerely, 

Alison Farrell, Ph.D. 

Senior Editor 

PLOS Medicine